# The performance of partially substituted composite ester materials with weathered red-bed soil in ecological restoration

Zhen Liu[1,2], Yongtao Wu[1,2], Jin Liao[1,2], Dexian Li[1,2], Cuiying Zhou[1,2]*

1 Guangdong Engineering Research Centre for Major Infrastructures Safety, School of Civil Engineering, Sun Yat-sen University, Guangzhou, China, 2 Guangzhou Engineering Research Centre for Major Infrastructures and Safety in Transporation, School of Civil Engineering, Sun Yat-sen University, Guangzhou, China

* zhoucy@mail.sysu.edu.cn

**Data Availability Statement:** The related data that support the findings of this study has been uploaded to the GitHub platform. The related data that support the findings of this study can be got from the Microsoft Office Excel file, named

## Abstract

Ester materials have become a significant topic in ecological restoration because of their degradability and lack of pollution. However, these artificial materials have issues such as high resource consumption and high cost. Therefore, finding a scientific substitute for ester materials is crucial to reduce costs. This study proposes the use of weathered red-bed soil to partially replace ester materials. Orthogonal coupled compounding and ecological effect tests were performed to analyze the soil improvement mechanism based on the mineral composition, soil structure, and electrical conductivity properties of the weathered red-bed soil. The experimental findings indicated that the soil modified using ester materials exhibited improved strength, water retention, and aeration owing to changes in the soil structure. Plant germination and height increased by 55% and 37 mm, respectively, when using a ratio of 15 g/m$^2$ absorbent ester material, 2.5 g/m$^2$ adhesive ester material, and 5% weathered red-bed soil. Through this approach, the amount of ester material to be used could be further reduced by 75%. The weathered red-bed soil offers improved ecological effects by altering the physical, mechanical, and hydraulic properties of the soil structure. This study presents a theoretical foundation for ecological conservation using weathered red-bed soil as a substitute for certain ester materials.

## Introduction

Ecological degradation is a major environmental issue worldwide [1]; it typically manifests as destruction of vegetation, reduced soil quality, degraded ecosystem functions, and environmental contamination [2, 3]. Ester materials, such as organic polymers, plant-based fibers, and glucose esters, have been demonstrated to enhance water retention, soil strength, structure, and vegetation growth [4–6]. However, because ester materials are artificial, their mass production can result in excessive resource consumption and high costs. Therefore, exploring partial substitution options for ester materials is important for the application and development of new eco-environmental materials. Natural materials are widely used in the oil industry owing to their abundance, low cost, and environmental friendliness. The use of natural

"minimal-data" (https://github.com/RockSYSU/ecological_restoration.git).

**Funding:** This research is supported by the National Natural Science Foundation of China (Grant Numbers: 42293354, 42293351, 42293355, 42277131, 41977230), this research work was tolerated and supported by CCCC FOURTH HARBOR ENGINEERING CO., LTD (Grant Number: 20187614071020007). These fundings are all awarded by the author Cuiying Zhou. We declare that the funder has no known competing financial interests or personal relationships that could have appeared to influence the work reported in this paper. The funders had no role in study design, data collection and analysis, decision to publish, or preparation of the manuscript.

**Competing interests:** The authors have declared that no competing interests exist.

materials instead of esters is critical for the green development of new ecological restoration technologies.

Substitute materials for ecological restoration esters are classified as physical, chemical or biological according to the method used [7]. Physical methods involve mechanically combining the substitute and ester materials to form a uniform mixture; lime, fly ash, cement, and slag silicate are the most common physical substitutes [8, 9]. Researchers have mainly focused on enhancing the physical and mechanical properties of soil using homogeneous mixtures of ester materials and physical substitutes. Substitutes of this type are more commonly used because of their ability to significantly enhance soil strength and cohesiveness [10], crack resistance [11], and stability [12, 13], as well as their ease of construction and low cost. However, disturbance, corrosion, and degradation of the soil are problems. In chemical methods, a chemical product generated by the polymerization of an ester with one or more chemical substitution materials is sprayed over the soil. Ions, lignin sulfonates, and calcium chloride are the main chemical substitutes [14, 15]. Chemical compounds bind soil particles, causing them to solidify inside soil pores, increasing soil strength [16, 17] and decreasing permeability [18, 19]. Compared with physical methods, chemical methods are generally more expensive and require sophisticated processes that disrupt the acid-base balance of the environment by considerably increasing the alkalinity of the soil. In biological methods, biological materials and esters are mixed uniformly, and the biological materials assist the esters in improving the soil structure. Biological enzymes, polymers, resins, and other materials are the most common biological substitution materials [20, 21]. By acting on soil particles, bio-substituted materials assisted by esters decrease the quantity of adsorbed water in the soil, creating an impermeable membrane, thus decreasing soil erosion [22] and permeability [23], and increasing soil swelling [24, 25]. However, an impermeable membrane requires a high water content, and a suitable ratio interval is difficult to obtain. Additionally, bio-substituted materials are challenging to extract, and excessive addition can negatively affect the environment. I In summary, most current ester substitution materials are artificial materials. Compared to the rock-weathered soil components found in nature, these components are more expensive, require more resources, are prone to secondary pollution, and require multiple treatments. Weathered red-bed soils are formed by the intense physicochemical weathering of red-bed rock bodies under hot and humid climatic conditions, and are widely distributed from the Early Proterozoic to the Neoproterozoic [26, 27]. Weathered red-bed soil has the same properties as ester materials, including strong cementation, hydrophilicity, and expandability. Weathered red-bed soil is environmentally friendly, widely distributed, and easily obtainable [28]. Exploring the possibility of partially replacing ester materials with red weathered soil is of great research significance for reducing costs and resource consumption.

In response to the above issues, this study conducted tests of physical, mechanical, and hydraulic properties, and ecological effects caused by the coupling and compounding resulting from the partial replacement of ester materials with natural red weathered soil. Furthermore, we investigated the process of soil improvement by connecting weathered red-bed soil and ester materials using the mineral composition, soil structure, and electrical properties of the weathered red-bed soil. We also investigated novel approaches and methodologies to provide new concepts and theoretical support for soil improvement research concerning weathered red-bed soil.

## Research content and methods

### Analysis of the relationship between composite ester materials and weathered red-bed soil in ecological restoration

**Characterization of the ecological restoration effect of composite ester materials.** Ester materials include water-absorbing and adhesive ester compounds. An adhesive ester

**Table 1. Studies on ecological restoration indices of ester materials.**

| Property | Experiment | Evaluation index |
|---|---|---|
| Mechanics | Scouring and shear strength | Erosion rate, cohesion, and internal friction angle [32, 33] |
| Physics | Contractility, expansion, pH, and conductivity | Shrinkage, expansion rate, pH, and conductivity [34, 35] |
| Water physical | Moisture retention and permeability | Moisture content, permeability coefficient [36, 37] |
| Ecological benefit | Plant growth | Plant height and germination rate [38, 39] |

compound mainly consists of a hydrolysis-resistant polyester with a high molecular weight and viscosity. This polyester is insoluble but has good dispersibility in water. The primary characteristics of water-absorbing ester compounds are high water absorption capacity and low water release volume contraction [29]. According to the group's past research and engineering experience, ester materials have good ecological restorative effects [30, 31], in terms of both the bonding of adhesive ester materials and the water retention of water-absorbent ester materials, both of which jointly improve the overall performance of the soil during the interaction process. Studies on ecological restoration indices using ester materials are listed in Table 1. The evaluation indicators for the soil improvement effect of ester materials listed in Table 1 mainly include the erosion rate, cohesion, internal friction angle, moisture content, permeability coefficient, shrinkage rate, pH, conductivity, expansion rate, plant height, and plant germination rate.

**Characteristic indices of weathered red-bed soil.** The weathered red-bed soil was crushed with a wooden hammer, sieved through a standard sieve, and mixed with water to form a slurry, the viscosity of which was measured using a viscometer (Table 2). Weathered red-bed soil has a high viscosity, and the mineral composition and grain gradation of the soil affect its viscosity. An X-ray polycrystalline diffractometer was used to determine the relative content of the weathered red-bed soil minerals utilized in this work. The results are displayed in Fig 1A, and the particle grading curves are shown in Fig 1B. The clay mineral content in weathered red-bed soil is high, and the main structures are composed of silicon-oxygen tetrahedra and aluminum or magnesium-oxygen-hydrogen octahedra arranged in a two-dimensional plane. These tetrahedral or octahedral units have strong polarity and are prone to ion exchange [40]. Additionally, the particle size and high specific surface energy of the clay minerals further improve their adsorption and ion exchange capabilities [41]. Thus, weathered red-bed soil exhibits high cementation, hydrophilicity, and expansibility.

**The substitution principle of weathered red-bed soil for composite ester materials.** Table 3 lists the features of weathered red-bed soil compared with those of ester materials, and

**Table 2. Viscosity of weathered red-bed soil.**

| Parameter type | value | | | | | | |
|---|---|---|---|---|---|---|---|
| Natural moisture content (%) | 21.52–29.83 | | | | | | |
| Density (g/cm$^3$) | 2.54–2.72 | | | | | | |
| Bulk density (g/cm$^3$) | 1.41–1.59 | | | | | | |
| Cohesion (kPa) | 20.4–29.3 | | | | | | |
| Internal friction angle (˚) | 18.0–22.6 | | | | | | |
| Permeability coefficient ($\times 10^{-7}$cm/s) | 4.25–6.72 | | | | | | |
| Particle size (mm) | <5.000 | <2.000 | <1.000 | <0.500 | <0.250 | <0.100 | <0.075 |
| Viscosity (Pa.s) | 13.000 | 38.000 | 44.000 | 60.000 | 67.000 | 59.000 | 68.000 |

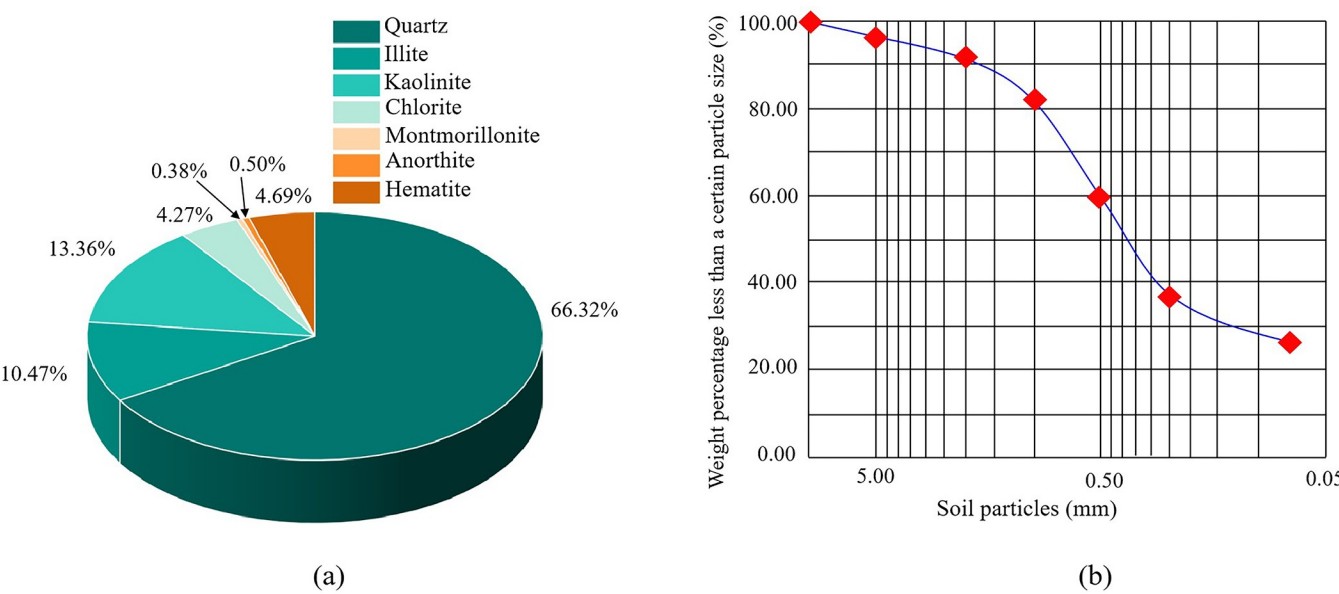

**Fig 1. Characteristics of weathered red-bed soil.** (a) Relative mineral content, (b) Grain size distribution curve.

shows that weathered red-bed soil and ester materials have similar properties, such as strong cementation, hydrophilicity, and expansibility. For ecological protection, strongly hydrophilic and expandable materials are added to soil to improve water retention, loosen the soil structure, and stimulate plant growth and development. The weathered red-bed soil volume can create a skeleton effect in the soil. After the evaporation of water, the soil structure becomes relatively stable. As ester materials lose water, their volume shrinks drastically, inducing particle rearrangement within the soil body, which considerably changes its strength. As a result, using weathered red-bed soil combined with ester materials can improve soil strength and stability during repeated water absorption-release processes, and prevent soil slumping caused by excessive ester material addition. Therefore, in this study, weathered red-bed soil was used as a partial substitute for ester materials, and coupled compounding and ecological effect tests were performed on the weathered red-bed soil and ester materials (Fig 2).

**Properties of weathered red-bed soil after partial substitution for composite ester materials.** *Compound test of ester materials.* Samples were collected from a highway slope in South

**Table 3. A comparison of the properties of weathered red-bed soil and ester materials.**

| Order number | Ester materials | Weathered red-bed soil |
|---|---|---|
| 1 | Strong water absorption: carboxyl and hydroxyl groups are prone to ion exchange [42]. | Strong water absorption: tetrahedron or octahedron particle arrangements are prone to ion exchange [43]. |
| 2 | Strong cementation: the polymer chain enhances the connection between particles [44]. | Strong cementation: clay minerals enhance the connection between particles [45]. |
| 3 | strong expansibility: the negative ion repulsion causes expansion of the network structure [46]. | Strong expansibility: the negative charge repulsion of clay minerals increases the number of pores in the soil [47]. |
| 4 | Without changing the soil structure, it adsorbs on the soil surface to form an elastic film [44]. | It does not react with the soil and adsorbs on the soil surface to form a hydration film [48]. |

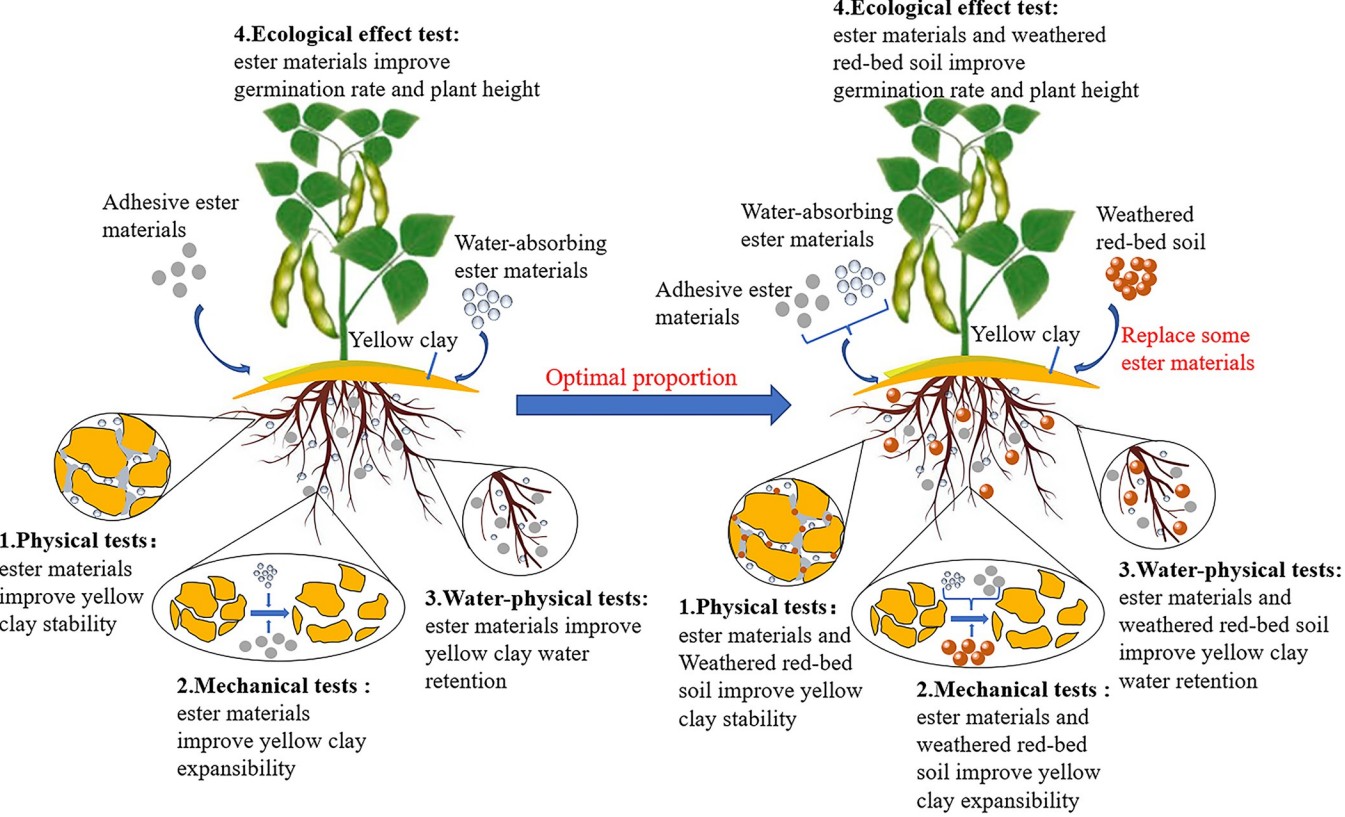

**Fig 2. Test idea diagram.**

China. This site has poor plant growth and is exposed throughout the year. The soil was Quaternary Holocene residual yellow clay, and its color was yellow-brown, which is typical of yellow clay in South China. Table 4 shows the basic properties of the yellow clay soil.

The purpose of the ester material compounding test was to investigate the effects of different ester material ratios on various factors, including plant germination rate, plant height, scour resistance, direct shear, water retention, shrinkage, swelling, infiltration, pH, and electrical conductivity. The aim of this test was to determine the optimal ratio under the applied conditions. Pigeon peas were used as the vegetation type. Pigeon peas can grow up to 3 m in height, have a root system depth of up to 3 m, are drought-resistant, grow and reproduce easily, and have rhizomes that promote nitrogen fixation. This species is commonly used in ecological restoration engineering [50]. The ratios are listed in Table 5.

Plant growth test steps:

1. Soil sample preparation: Yellow clay was weighed into the soil tray to equal the ratio in Table 5, after which the corresponding mass of weathered red-bed soil and adhesive ester material was weighed into the soil tray and mixed well. Next, 500 g of the well-mixed soil sample was weighed and placed in a Petri dish. As illustrated in Fig 3, three sets of each proportion were tested.

**Table 4. Basic parameters of yellow clay soil samples(data were obtained from [49]).**

| Natural moisture content (%) | Natural density (g/cm³) | Dry density (g/cm³) | Liquid limit (%) | Plastic limit (%) | Viscosity (Pa·s) |
| --- | --- | --- | --- | --- | --- |
| 25.26 | 1.52 | 1.21 | 54.10 | 36.10 | 29.00 |

**Table 5. Ratios of ester materials for the plant growth compound test.**

| Group number | Water-absorbing ester material (g/cm²) | Adhesive ester material (g/cm²) |
|---|---|---|
| 1 | 60.00 | 15.00 |
| 2 | 60.00 | 10.00 |
| 3 | 60.00 | 5.00 |
| 4 | 30.00 | 15.00 |
| 5 | 30.00 | 10.00 |
| 6 | 30.00 | 5.00 |
| 7 | 15.00 | 15.00 |
| 8 | 15.00 | 10.00 |
| 9 | 15.00 | 5.00 |
| 10 | 0.00 | 0.00 |

2. Planting: After the pigeon pea seeds were immersed in water for germination, they were placed in Petri dishes containing soil samples and topped with soil samples at a depth of 1 cm, with 30 seeds sown in each pot.

3. Spraying of adhesive ester: According to the ratios in Table 5, the corresponding quantities of adhesive ester materials were weighed and 200 mL of water was added to generate a uniformly sprayed dispersion over the surface of the Petri dish. The Petri dish was placed in a well-lit area and watered every three days to maintain plant growth. The amount of water applied each time was consistent for each Petri dish.

4. Measurement data collection: Every day following seed germination, the number of germinated plants was counted, and their height was measured. After counting the germination of one plant, the seedling germination height reached 1 cm. When measuring plant height, 3–5 plants were measured in the same pot, and the average height was taken as the plant height of the group of plants on that day.

5. Calculation: Calculate the germination rate using Eq (1):

$$\chi = \frac{a_1}{a_0} \times 100\% \tag{1}$$

Where $\chi$ denotes the germination rate, $a_1$ is the number of germinated plants, and $a_0$ is the total number of seeds for testing.

Coupled composite test procedure: Sieved yellow clay was placed in a soil tray, and weathered red-bed soil, water-absorbing ester material, and water were added, stirred well to form mud, and smoothed. The adhesive ester material was then mixed with water to create a dispersion solution that was evenly sprayed on the surface of the mud.

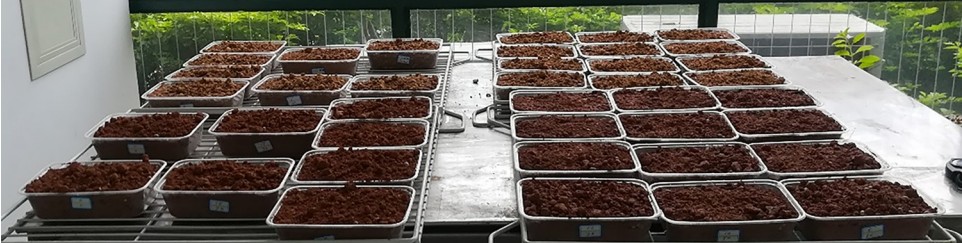

**Fig 3. Culture plate and soil samples.**

(1) For scouring resistance, the specimens were maintained at room temperature for 24 h before testing. The tilt angle of the dirt tray was set to 50° based on the slope gradient of the project. The scouring water flow rate was set to 1 L/min, and the scouring time was set to 10 min based on the local rainfall characteristics. The soil tray was allowed to stand for 1 h after the scouring. After removing the water, the residual silt sand was weighed and baked at 105°C for 24 h. Eq (2) was used to compute the erosion rate:

$$\lambda = \frac{m_0}{m_1 + m_2 + m_3} \times 100\% \tag{2}$$

where $\lambda$ is the erosion rate, $m_0$ is the mass of sediment washed out (g), $m_1$ is the mass of clay (g), $m_2$ is the mass of red-layered weathered soil added (g), and $m_3$ is the mass of ester material added (g).

(2) For maintenance, the specimens were wrapped in plastic wrap and kept in a humidifier for 24 h prior to shear testing. The shear rate was set at 0.8 mm/min, and the maximum shear displacement for each 0.8 mm reading was 6 mm. Vertical pressures of 50, 100, 200, and 400 kPa were applied to each group of specimens, and the cohesion and internal friction angles of the specimens were calculated based on the intersection and slope of the curves.

(3) After placing the specimens in an incubator with constant temperature and humidity, water retention, shrinkage, pH, and conductivity tests were performed. The inner diameter of the ring knife was 61.8 mm. Its height was 40 mm and the total weight of the test specimens was measured using a balance every 12 h. The following formula was used to compute the water content at various points:

$$\omega = \frac{m_O - (m_d + m_c + m_b)}{m_d - (m_c + m_b)} \times 100\% \tag{3}$$

where $\omega$ is the water content, $m_O$ is the total mass of the specimen (g), $m_d$ is the dry weight of the specimen (g), $m_c$ is the mass of the ring knife (g), and $m_b$ is the mass of the bottom cover (g).

(4) After conducting the water retention test, the specimen was dried, and its volume was measured and compared with the original volume to calculate the shrinkage rate using the following formula:

$$\gamma = \frac{V_0 - \left(\frac{M_2 - M_1}{\rho_{wT}} - \frac{M_3 - M_0}{\rho_n}\right)}{V_0} \times 100\% \tag{4}$$

where $M_0$ is the initial mass of the specimen (g), $M_1$ is the initial reading of the balance (g), $M_2$ is the initial reading of the balance when the specimen was submerged (g), $M_3$ is the mass of the wax-sealed specimen (g), $\rho_{wT}$ is the density of pure water at $T°C$ (g/cm$^3$), $\rho_n$ is the density of wax (g/cm$^3$), $\gamma$ is the shrinkage rate, $V_0$ is the initial volume of the specimen (cm$^3$), and $V_1$ is the volume of the specimen after shrinkage (cm$^3$).

(5) During the expansion rate test, constantly add pure water so that the water level in the tank is never higher than the upper surface of the specimen and the specimen is fully enlarged. Record the end percentage meter reading 24 hours before the test concludes, when the hourly deformation of the specimen is less than 0.01mm. Calculate the expansion rate using the following formula:

(5) During the expansion rate test, pure water was constantly added to ensure that the water level in the tank was never higher than the upper surface of the specimen and that the specimen was fully enlarged. The end percentage meter reading was recorded 24 h before the test was concluded when the hourly deformation of the specimen was less than 0.01 mm. The expansion rate was calculated using the following formula:

$$v = \frac{h_2 - h_1}{H} \times 100\% \qquad (5)$$

where $v$ is the expansion rate, $h_1$ is the initial meter reading (mm), $h_2$ is the final meter reading (mm), and $H$ is the height of the ring cutter (mm).

(6) The permeability of the specimens was tested under two spatial constraints: annular and upper- and lower-end constraints during expansion, as shown in Fig 4. The saturated specimen was inserted into a variable head permeability vessel, tightened with a nut, and sealed until it was resistant to water and air leakage. The variable head pipe was injected with pure water, the water was passed through the specimen, and the starting and ending head heights of the variable head pipe, along with the related time intervals, were measured and recorded. The test was repeated 5–6 times at different starting head heights. The coefficient was calculated as follows:

$$K_T = 2.3 \frac{aL}{At} log \frac{H_1}{H_2} \qquad (6)$$

where $K_T$ is the permeability coefficient (cm/s), a is the cross-sectional area of the variable head pipe (cm$^2$), A is the cross-sectional area of the ring knife (cm$^2$), L is the seepage diameter (cm), t is the time interval between the measurement of the starting and ending heads (t), $H_1$ is the starting head height (cm), and $H_2$ is the ending head height (cm).

The three groups of tests were averaged, and the results are shown in Fig 5. Fig 5(A) demonstrates that the Group 2 trials had the highest germination rate and plant height, as well as the lowest erosion rate and highest coefficient of permeability. Therefore, when ester materials were applied alone, the best ratios were 60 g/m$^2$ for water-absorbent esters and 10 g/m$^2$ for adhesive ester materials.

**Coupled compounding test of weathered red-bed soil and ester materials.** A coupled compounding test of weathered red-bed soil and ester materials was performed to obtain the

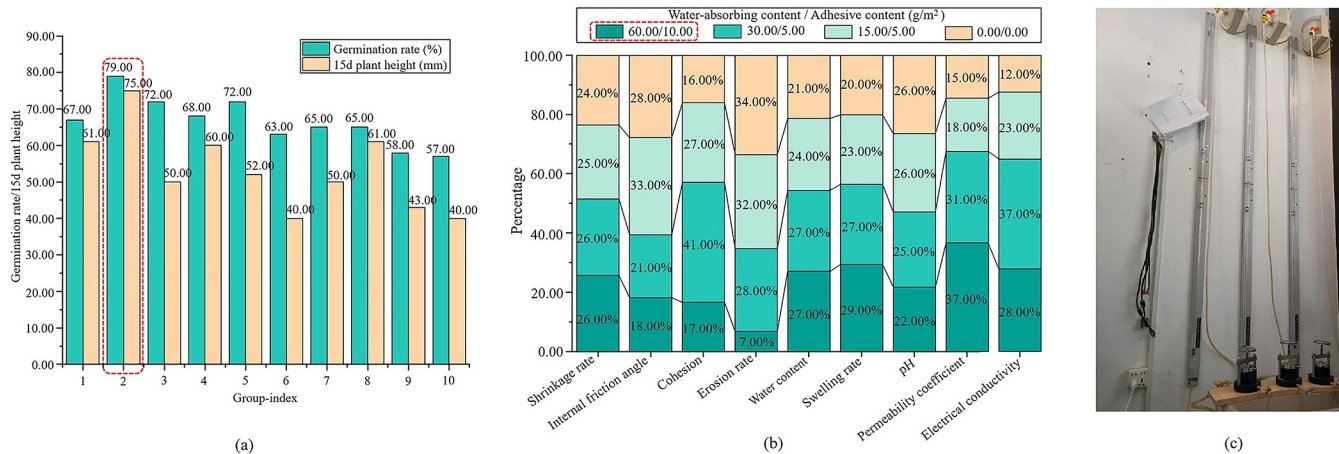

**Fig 4. Permeability test.** (a) TST-55 penetrometer, (b) Specimen subjected to upper and lower limits, (c) Sample not subject to the upper and lower limits.

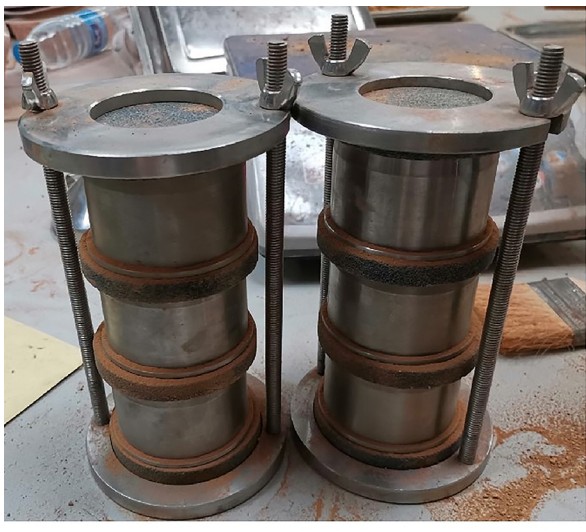
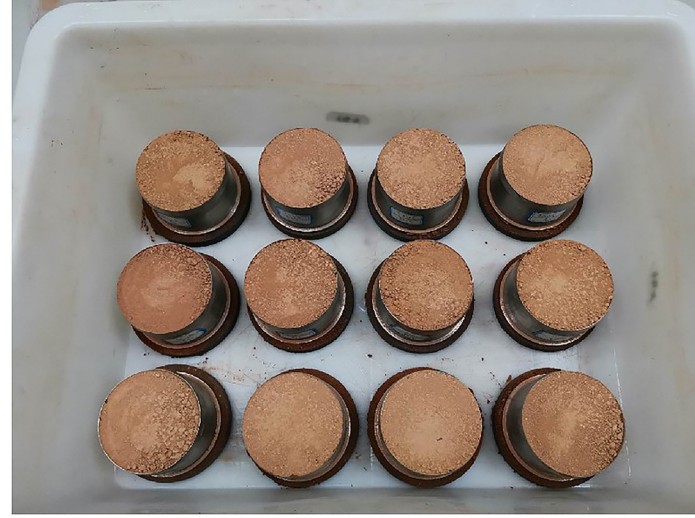

(a)                                                                    (b)

**Fig 5. The effect of ester materials alone.** (a) Plant growths, (b) Soil parameters.

ecological restoration indices listed in Table 1 to evaluate the ecological protection effect of weathered red-bed soil. An orthogonal test with three selected components and four levels (1, 2, 3, and 4) was created based on the optimal proportions of the applied ester materials. The orthogonal sequences of material ratios are listed in Table 6. Among them, the water-absorbing and adhesive ester materials were calculated as the amount added per square meter of 10-cm-thick yellow clay. For weathered red-bed soil, the amount added was calculated as its percentage of the mass of yellow clay. Changes in the mechanical, hydraulic, and physical properties of the soil were then investigated. To validate the feasibility of weathered red-bed soil as an ecological protection material, we performed eight tests, including scour resistance, direct shear, water retention, shrinkage, swelling, infiltration, and pH tests. Thirty sample sets were used in each test, and three replicates were prepared for each set of specimens. The test procedures and data collection were the same as those described in the previous section to confirm the feasibility of using weathered red-bed soil as an ecological protection material.

**Ecological effect test of coupling effect between weathered red-bed soil and ester materials.** To investigate the optimal ratios for the coupling effect of weathered red-bed soil and ester materials, the amount of ester material was reduced based on the optimal ester material ratio, and weathered red-bed soil was added to investigate its effect on plant growth. Table 7 presents the orthogonal sequences of a three-factor, three-level orthogonal test created using the same factors as in the previous section. The test techniques and data collection were the same as in the previous section, except that the red-layer weathered soils were partially substituted for the ester materials in the ratios listed in Table 7.

## Results and discussion

### Properties of the weathered red-bed soil partially substituted with composite ester materials

**The change in erosion rate of modified soil.** Fig 6 shows a graph comparing the erosion rate based on the findings of the scour resistance test. As shown, the erosion rate of each group

**Table 6. Ratios of ester materials to weathered red-bed soil.**

| Group number | Factors | | |
|---|---|---|---|
| | $p(g/m^2)$ | $a(g/m^2)$ | r (%) |
| 1 | $p_1$:0.00 | $a_1$:0.00 | $r_1$:0.00 |
| 2 | $p_2$:15.00 | a2:2.50 | $r_1$:0.00 |
| 3 | $p_2$:15.00 | $a_3$:5.00 | $r_1$:0.00 |
| 4 | $p_3$:30.00 | $a_2$:2.50 | $r_1$:0.00 |
| 5 | $p_3$:30.00 | $a_3$:5.00 | $r_1$:0.00 |
| 6 | $p_4$:60.00 | $a_4$:30.00 | $r_1$:0.00 |
| 7 | $p_1$:0.00 | $a_1$:0.00 | $r_2$:1.00 |
| 8 | $p_2$:15.00 | $a_2$:2.50 | $r_2$:1.00 |
| 9 | $p_2$:15.00 | $a_3$:5.00 | $r_2$:1.00 |
| 10 | $p_3$:30.00 | $a_2$:2.50 | $r_2$:1.00 |
| 11 | $p_3$:30.00 | $a_3$:5.00 | $r_2$:1.00 |
| 12 | $p_4$:60.00 | $a_4$:30.00 | $r_2$:1.00 |
| 13 | $p_1$:0.00 | $a_1$:0.00 | $r_3$:2.50 |
| 14 | $p_2$:15.00 | $a_2$:2.50 | $r_3$:2.50 |
| 15 | $p_2$:15.00 | $a_3$:5.00 | $r_3$:2.50 |
| 16 | $p_3$:30.00 | $a_2$:2.50 | $r_3$:2.50 |
| 17 | $p_3$:30.00 | $a_3$:5.00 | $r_3$:2.50 |
| 18 | $p_4$:60.00 | $a_4$:30.00 | $r_3$:2.50 |
| 19 | $p_1$:0.00 | $a_1$:0.00 | $r_4$:5.00 |
| 20 | $p_2$:15.00 | $a_2$:2.50 | $r_4$:5.00 |
| 21 | $p_2$:15.00 | $a_3$:5.00 | $r_4$:5.00 |
| 22 | $p_3$:30.00 | $a_2$:2.500 | $r_4$:5.00 |
| 23 | $p_3$:30.00 | $a_3$:5.00 | $r_4$:5.00 |
| 24 | $p_4$:60.00 | $a_4$:30.00 | $r_4$:5.00 |
| 25 | $p_1$:0.00 | $a_1$:0.00 | $r_5$:10.00 |
| 26 | $p_2$:15.00 | $a_2$:2.50 | $r_5$:10.00 |
| 27 | $p_2$:15.00 | $a_3$:5.00 | $r_5$:10.00 |
| 28 | $p_3$:30.00 | $a_2$:2.50 | $r_5$:10.00 |
| 29 | $p_3$:30.00 | $a_3$:5.00 | $r_5$:10.00 |
| 30 | $p_4$:60.00 | $a_4$:30.00 | $r_5$:10.00 |

Notes: p: adhesive ester material, a: water-absorbing ester material, r: weathered red-bed soil

decreased as the amount of weathered red-bed soil increased. The addition of weathered red-bed soil lowered the erosion rate by 0.5–12.5%. Therefore, the addition of weathered red-bed soil improved the erosion resistance of the soil. Furthermore, the soil erosion rate remained low as the amount of ester material decreased and the amount of weathered red-bed soil increased. This indicates that weathered red-bed soil can be used as a partial substitute for ester materials to reduce their use in soil erosion resistance projects.

**Variation of cohesion and internal friction angle of modified soil.** As shown in Figs 7 and 8, the cohesiveness and internal friction angle of the specimens were plotted based on the shear strength test results. According to Fig 7, the addition of weathered red-bed soil enhanced the cohesiveness of the specimens. Their cohesive force increased from 1.29 to 21.51 kPa after the weathered red-bed soil was applied, compared to the soil with only ester materials. The specimens with 10% added weathered red-bed soil, 15 g/m² water-absorbing ester material, and 2.5 g/m² adhesive ester material exhibited the maximum cohesion value of 40.3 kPa. As

**Table 7. Orthogonal test group ratios for plant growth measurements.**

| Group number | Factors | | |
|---|---|---|---|
| | P (g/m$^2$) | A (g/m$^2$) | R (%) |
| 1 | P$_1$: 30.00 | A$_1$: 5.00 | R$_1$: 1.00 |
| 2 | P$_1$: 30.00 | A$_2$: 2.50 | R$_2$: 2.50 |
| 3 | P$_1$: 30.00 | A$_3$: 0.00 | R$_3$: 5.00 |
| 4 | P$_2$: 15.00 | A$_1$: 5.00 | R$_2$: 2.50 |
| 5 | P$_2$: 15.00 | A$_2$: 2.50 | R$_3$: 5.00 |
| 6 | P$_2$: 15.00 | A$_3$: 0.00 | R$_1$: 1.00 |
| 7 | P$_3$: 0.00 | A$_1$: 5.00 | R$_2$: 2.50 |
| 8 | P$_3$: 0.00 | A$_2$: 2.50 | R$_1$: 1.00 |
| 9 | P$_3$: 0.00 | A$_3$: 0.00 | R$_3$: 5.00 |
| 10 | P$_4$: 60.00 | A$_4$: 10.00 | R$_4$: 0.00 |
| 11 | P$_3$: 0.00 | A$_3$: 0.00 | R$_4$: 0.00 |

Notes: P: adhesive ester material, A: water-absorbing ester material, R: weathered red-bed soil

shown in Fig 8, the addition of weathered red-bed soil had little influence on the internal friction angle of the specimens. This indicates that the increase in shear strength of the soil conferred by the addition of weathered red-bed soil changed only the cohesion of the soil, not the angle of internal friction.

**Change in moisture content, shrinkage rate, pH, and electrical conductivity of modified soil.**   Temporal changes in the water content of the specimens were plotted based on the results of the water retention tests (Fig 9). The addition of weathered red-bed soil enhanced the water content of the specimens. To better study the variation in the moisture content of the samples, the moisture content of the samples with different amounts of weathered red-bed soil added at 120 h was plotted, as shown in Fig 10. As shown, the addition of weathered red-bed soil to ester materials significantly increased the water content, with a maximum increase of 9.3%. This indicates that the addition of weathered red-bed soil improved the water retention capacity of the soil and increased plant growth.

Fig 11 shows the shrinkage rates of the specimens. The shrinkage rate of the specimens decreased as the amount of weathered red-bed soil increased. The addition of weathered red-bed soil reduced specimen shrinkage by 0.18%–5.42% compared with the control group (only ester materials added). This indicates that weathered red-bed soil positively affected soil shrinkage and prevented soil slumping.

The fluctuation patterns of the pH and electrical conductivity values were plotted independently (Figs 12 and 13). As shown in Fig 12, the pH values of the groups did not change consistently with the addition of weathered red-bed soil. Therefore, the weathered red-bed soil had little effect on the pH of the different samples. As shown in Fig 13, the conductivity values of all specimens increased as the amount of weathered red-bed soil increased. The application of weathered red-bed soil enhanced conductivity by 0.25%–118% compared to the control group. This indicates that the application of weathered red-bed soil in water-soluble salts could provide more mineral nutrients to plants.

**Change in expansion rate of modified soil.**   The changes in expansion rate were plotted based on the swelling test results (Fig 14). The expansion rates of all specimens showed an upward trend with the addition of weathered red-bed soil. The expansion rates of the specimens increased by 0%–2.42% compared to that of the control. The sample with 5% additional weathered red-bed soil exhibited the greatest increase in expansion rate. The expansion rate increased by 3.62% compared with that of the control. This indicates the addition of

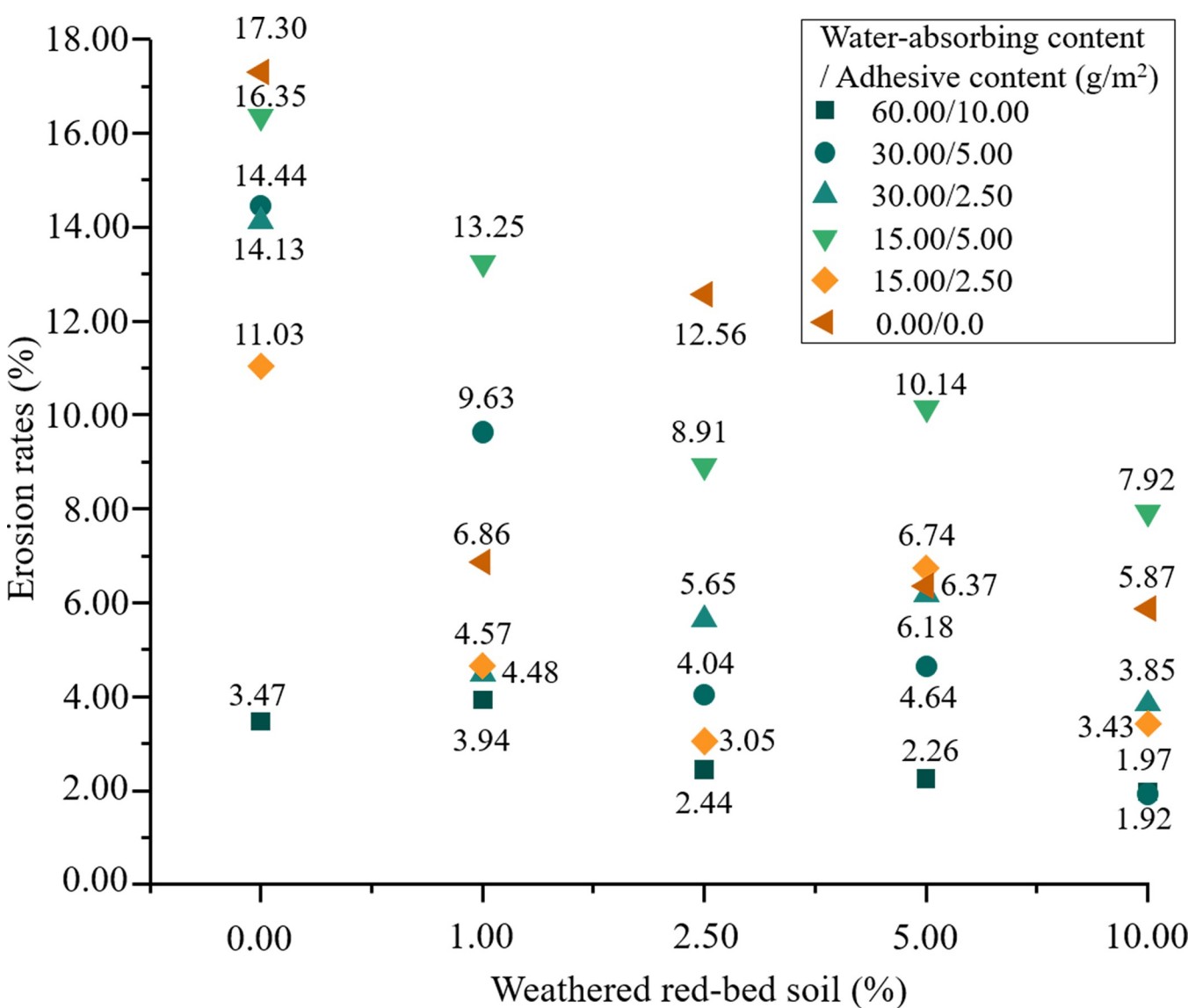

**Fig 6. Comparison of erosion rate.**

weathered red-bed soil resulted in higher clay particle content and stronger mineral hydrophilicity in the specimens, which are beneficial for plant root ion uptake and assist in ecological protection.

**Variation of permeability coefficient in modified soil.** The permeability coefficients of each specimen were plotted based on the permeation test results (Fig 15). When the specimens were confined by the ring direction and the upper and lower ends during expansion, the permeability coefficients decreased with the addition of weathered red-bed soil (Fig 15A). Compared with the control, the specimens with additional weathered red-bed soil exhibited a reduction in the permeability coefficient of 0.8%–72.7%. The permeability coefficients of the specimens were limited only by the ring direction, without the top and bottom ends, and increased by 7.9%–152.5% with the addition of weathered red soil (Fig 15B) compared with the control. When the upper and lower limitations were applied, the permeability coefficient decreased, whereas when these limitations were not applied, the permeability coefficient

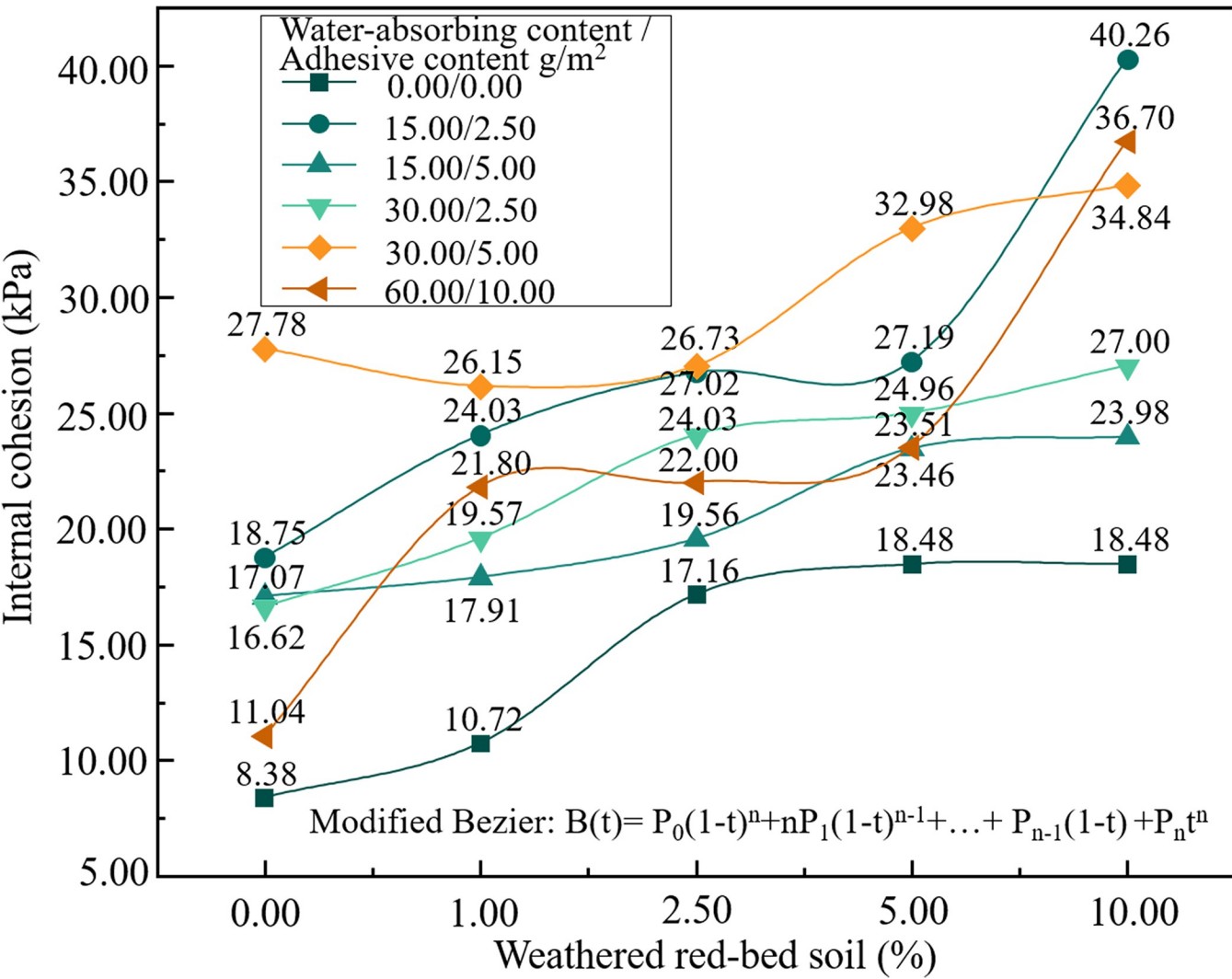

**Fig 7. Comparison of the cohesive forces of each sample.**

increased. This indicates that the addition of weathered red-bed soil improved the vertical water flow in the soil, which is beneficial for plant water absorption.

**Variation in plant height and germination rate.** The growth states of wood beans and germination rate plotted against plant height are shown in Figs 16 and 17, respectively. As shown in Fig 17, group 11 had the lowest germination rate and plant height; thus, group 11 was used as a control. Compared with the control, the optimal ratio of ester materials boosted the germination rate by 10% and plant height by 3 mm in group 10. Germination rates increased by 5%–45% and plant height increased by 4–34 mm in all experimental groups with the addition of weathered red-bed soil combined with ester materials compared with group 10, whereas germination rates increased by 15%–55% and plant height increased by 7–37 mm in group 11. This indicates that the plant germination rate and height increased with the addition of weathered red-bed soil. Group 5 had the best ratio of weathered red-bed soil to ester material: 15 g/m$^2$ water-absorbing ester material, 2.5 g/m$^2$ adhesive ester material, and 5% weathered red-bed soil. The amount of ester materials can be reduced by 75% using this ratio, and the ecological benefits of weathered red-bed soil have been demonstrated.

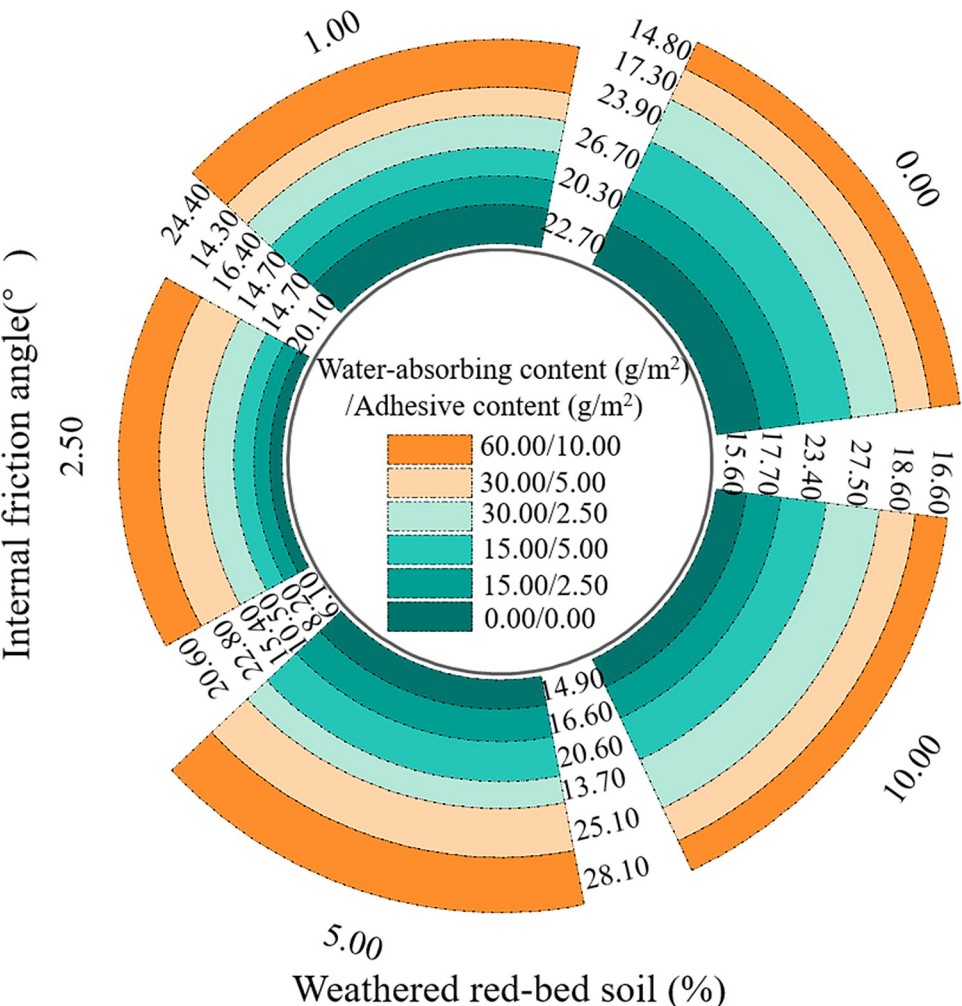

**Fig 8. Comparison of the internal friction angles of each sample.**

### The mechanism of weathered red-bed soil partially used as a partial substitute for composite ester materials

**Coupling mechanism between weathered red-bed soil and ester materials.** Ester materials produce stable linkages between soil particles via the elastic modes formed by polymer chains, boosting soil strength. Nevertheless, excessive application is likely to promote soil sloughing. The weathered red-bed soil contains a high concentration of clay minerals, which can strengthen the structure of the soil body and increase water absorption. As a result, by reducing the amount of ester material applied, the soil body can still be manufactured to have higher water and moisture retention performance due to the coupling effect of the three components. The addition of red weathered soil can not only reduce the usage of high-performance ester binder without reducing the soil body's strength and scour resistance, but also reduce the soil consolidation induced by high-performance ester binders.

**Mechanism of improvement of mechanical properties of soil using combined weathered red-bed soil and ester materials.** The erosion rate was reduced by 0.5%–12.5% after the partial substitution of ester materials with weathered red-bed soil in the test. This is because the weathered red-bed soil has high electrical conductivity and can easily absorb water. Therefore,

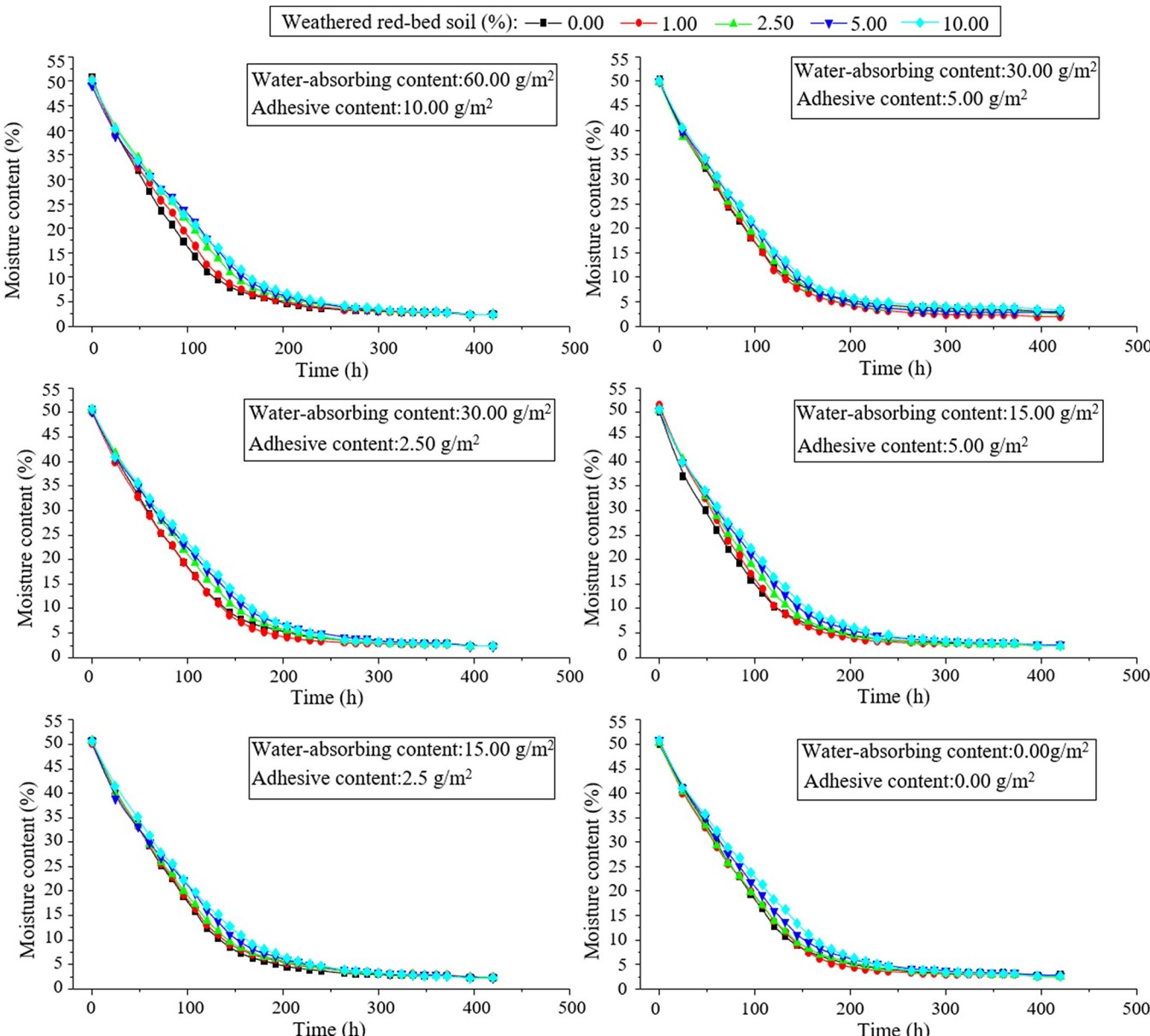

**Fig 9. Line graphs showing the temporal change in moisture content of different samples.**

a hydrated coating with strong adhesion formed on the surface of the clay mineral particles (Fig 18B), and these particles clung to the yellow clay particles [51, 52]. Owing to the considerable quantity of water adsorbed by the weathered red-bed soil, the hydration film on the surface of the yellow clay particles became thinner compared to the hydration film of the yellow clay particles shown in Fig 18(A). This resulted in smaller distances between the yellow clay particles, as well as increased particle bonding. Combining this effect with fact that the polymer chains of ester materials contain active groups, they link with each other and the soil particles via hydrogen bonding, intermolecular forces, and ion exchange; thus, the polymer chain entangles loose soil particles into a whole. As a result, there was an increase in the ability of the soil to withstand gravity and rainwater scouring damage, that is, scouring resistance. In

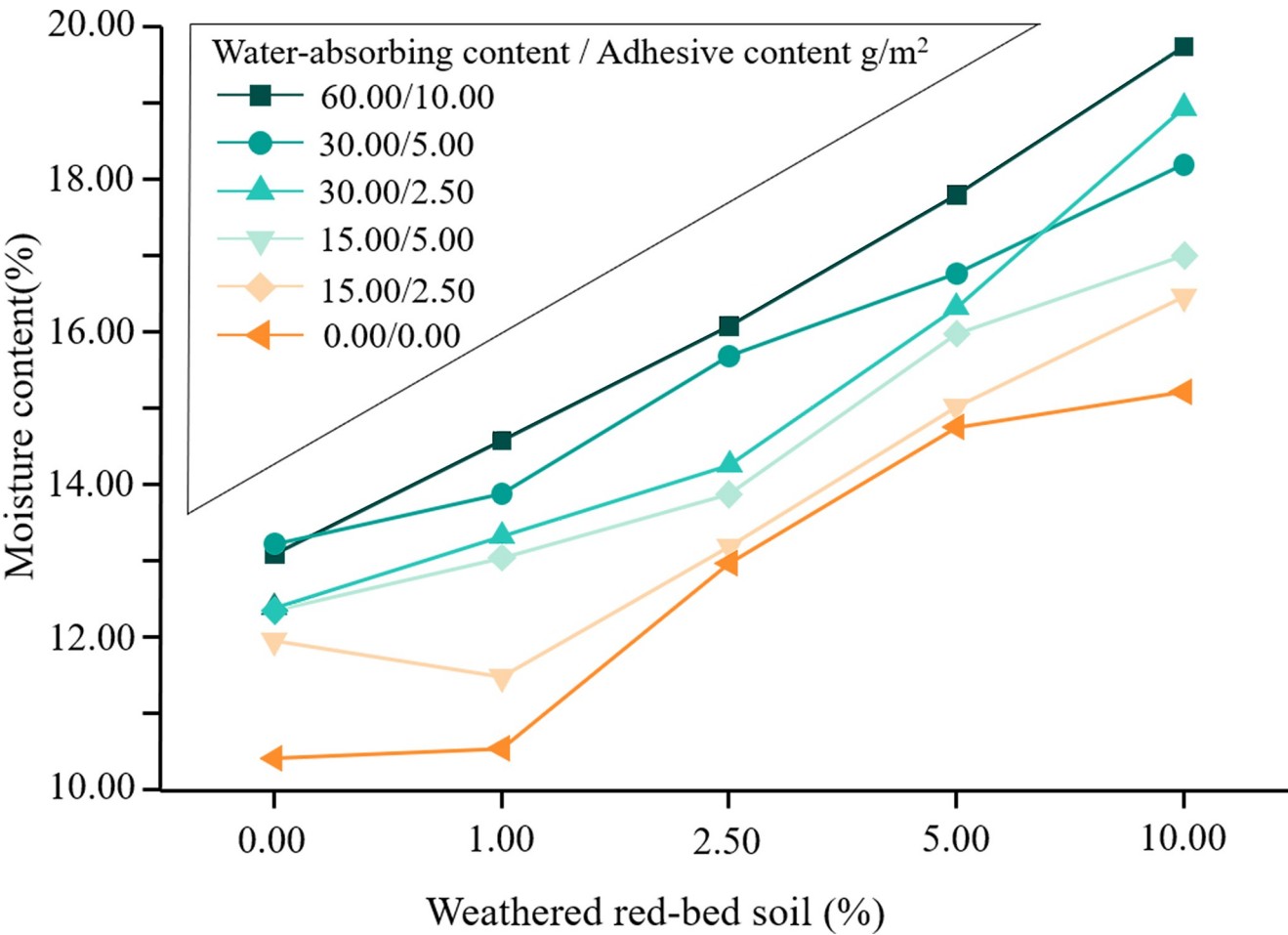

**Fig 10. Moisture content of each sample with different amounts of additional weathered red-bed soil over 120 h.**

addition, the cohesion of yellow clay increased by 1.29 kPa to 21.51 kPa. This is because the particle size of the weathered red-bed soil was much smaller, which increased the total specific surface area of the soil; it also increased the contact area of the particles within the soil, which increases the attraction between the particles. The addition of weathered red-bed soil increased cohesion more than the addition of ester materials alone. This shows that reducing the amount of ester materials used is more effective in improving the soil strength when weathered red-bed soils and ester materials are combined. The addition of weathered red-bed soil did not affect the internal friction angle of the soil samples because it did not modify the morphology of the yellow clay particles. Thus, weathered red-bed soil mainly improves yellow clay soil by decreasing the thickness of the hydration film and interparticle contact area, improving the mechanical properties of the yellow clay soil.

**The mechanism of improvement in chemical properties of weathered red-bed soil combined with ester materials.** As previously described, the surface of weathered red-bed soil particles has a negative charge; thus, the clay mineral particles attract cations such as $Na^+$, $K^+$, $Ca^{2+}$, and $NH^{4+}$ in water (Fig 19). The electrical conductivity tests performed on the soil revealed that conductivity tended to increase with the addition of weathered red-bed soil, increasing by 0.25%–118%. This was due to the increased attraction between a greater number of cations via cation exchange. In contrast, many cations were adsorbed during the production

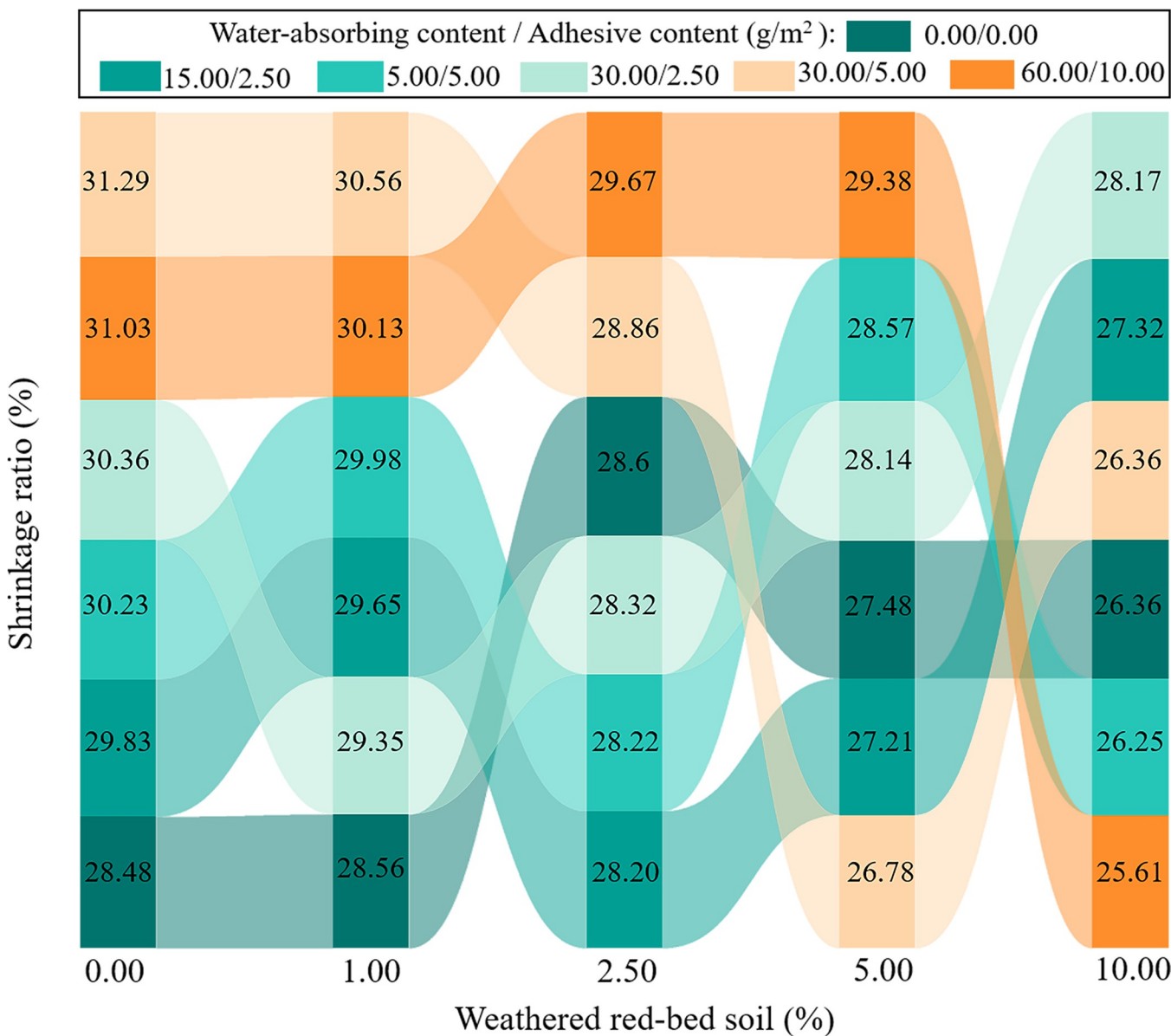

**Fig 11. Comparison of shrinkage rate.**

of weathered red-bed soil. These cations were carried to the soil body of the weathered red-bed soil, increasing the total quantity of cations; this, in turn, increased the electrical conductivity of the soil, allowing plants to receive adequate nutrient components and grow more effectively.

**Mechanism of improvement in soil–water physical properties with combined weathered red-bed soil and ester materials.** The water content increased by up to 9.3% when the ester material was partially substituted with weathered red-bed soil. This is because the water molecules absorbed sufficient energy to cross the potential barrier and break free from the clay mineral particles, resulting in an increase in the soil's water retention. The negative charge on the surface of the clay mineral particles caused them to repel each other. The shrinkage of yellow clay decreased by 0.18%–5.42% and the swelling increased by 0%–2.42%. The primary cause

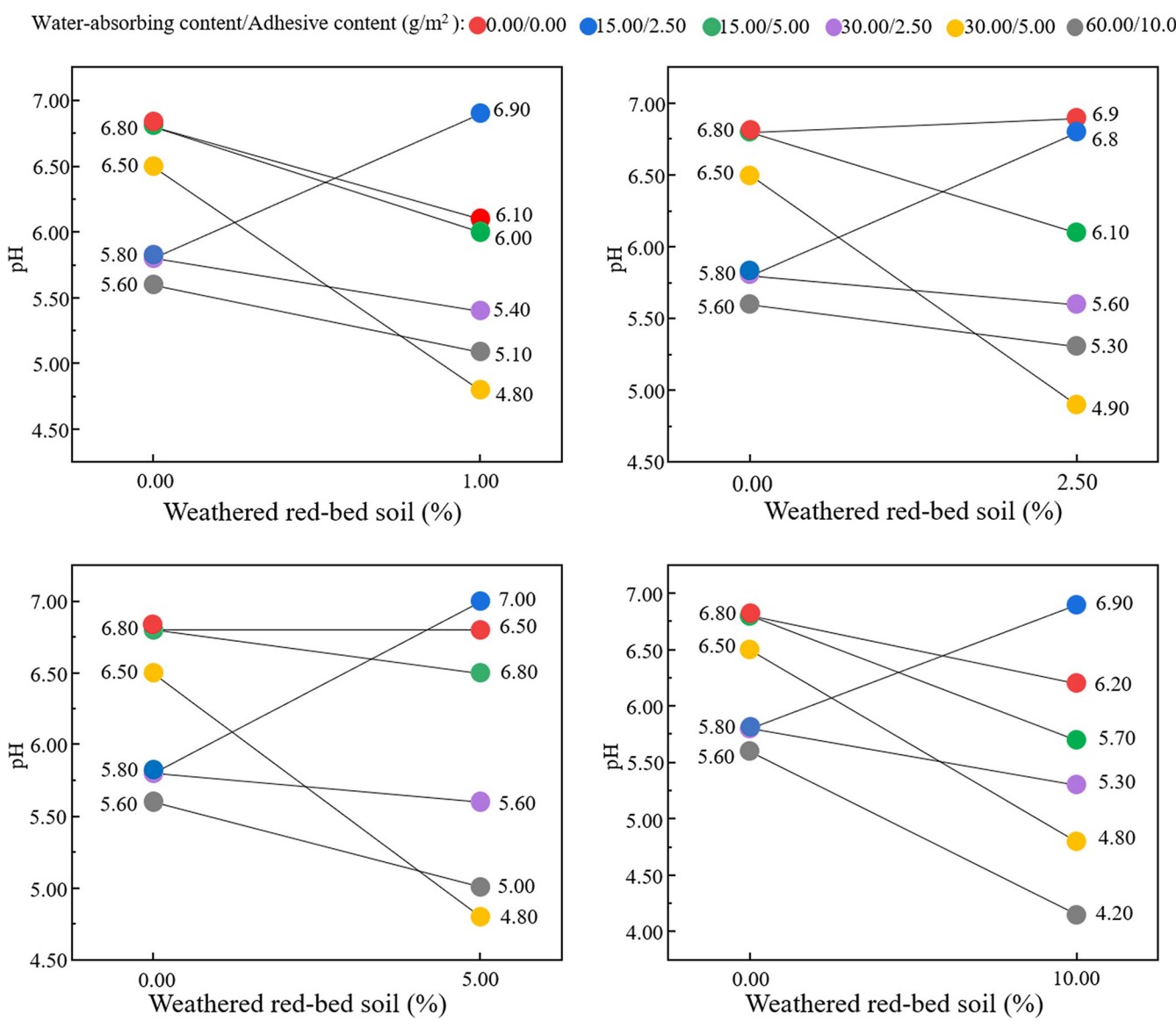

**Fig 12. pH change for different amounts of added weathered red-bed soils.**

was a change in the thickness of the hydration film on the surface of the soil particles. After the weathered red-bed soil in the soil body absorbed a substantial amount of water, the hydration film thickened dramatically, increasing the volume of the soil body, thereby enhancing expansion. However, the surface water loss contraction of soil particles, due to mutual repulsion [53], can maintain a certain distance and thus act as a skeleton, inhibiting the relative movement of soil particles, thus reducing the soil body shrinkage rate. In the permeability test, when the upper and lower limits were not applied to the soil samples, the permeability coefficients were reduced by 0.8%–72.7%. The soil particles had sufficient area to swell after water absorption. Simultaneously, during the swelling process, the size, number, and connectivity of pores in the soil increased. Thus, the permeability coefficients of the soil samples increased with the amount of weathered red-bed soil. After applying the upper and lower limits, the permeability coefficients increased by 7.9%–152.5%. A closed space was established inside the

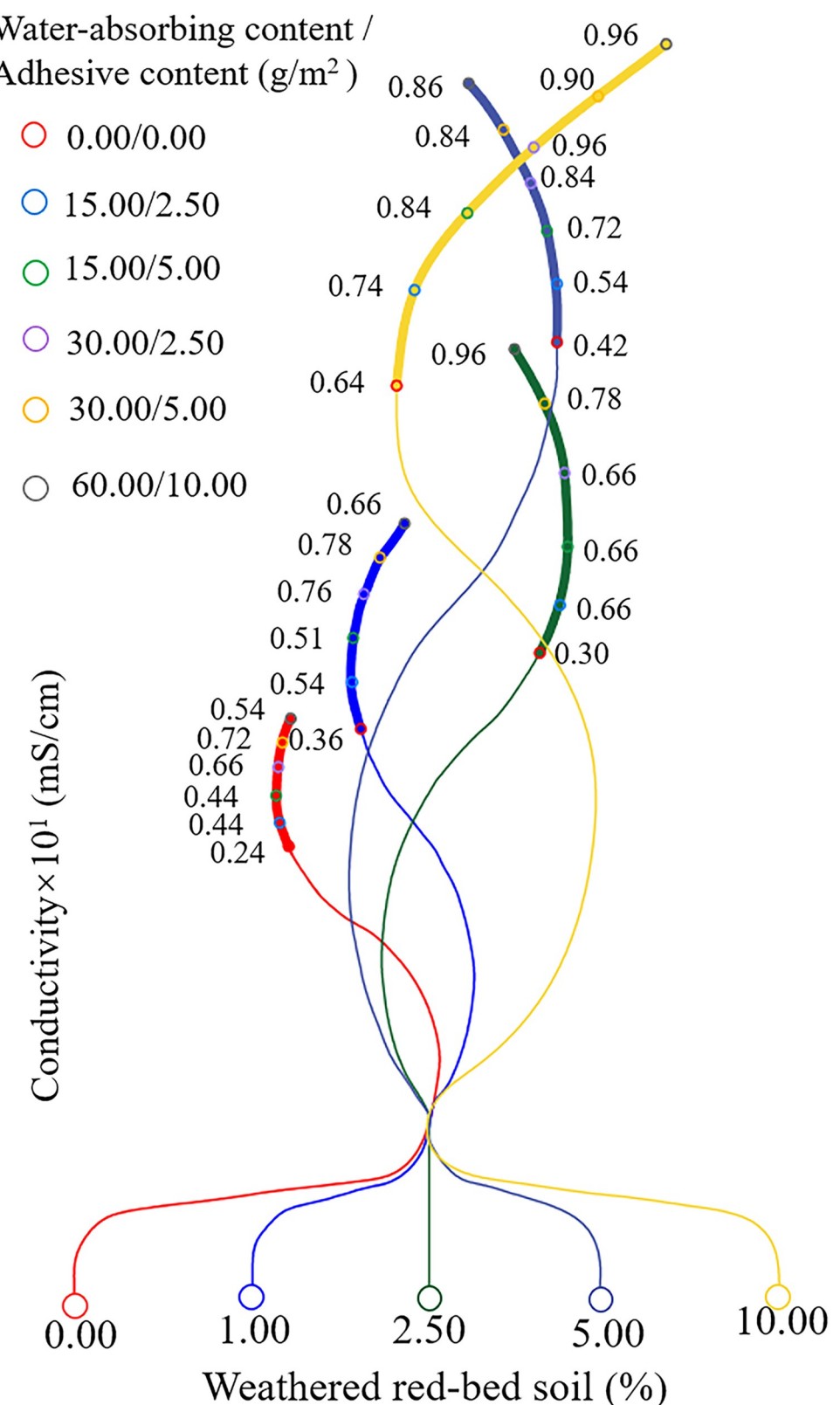

**Fig 13. Change in electrical conductivity for different amounts of added weathered red-bed soils.**

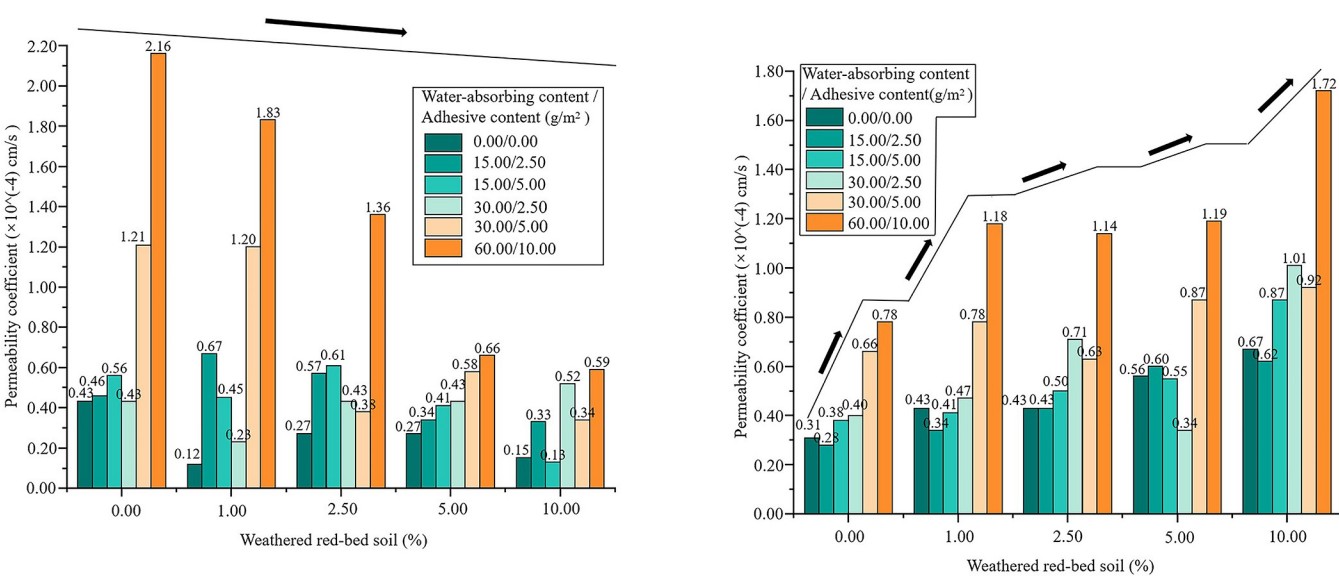

**Fig 14. Comparison of the expansion rate at different levels of added weathered red-bed soil.**

**Fig 15. Change in permeability coefficient at different levels of added weathered red-bed soil.** (a) Upper and lower limitations applied, (b) Upper and lower limitations not applied.

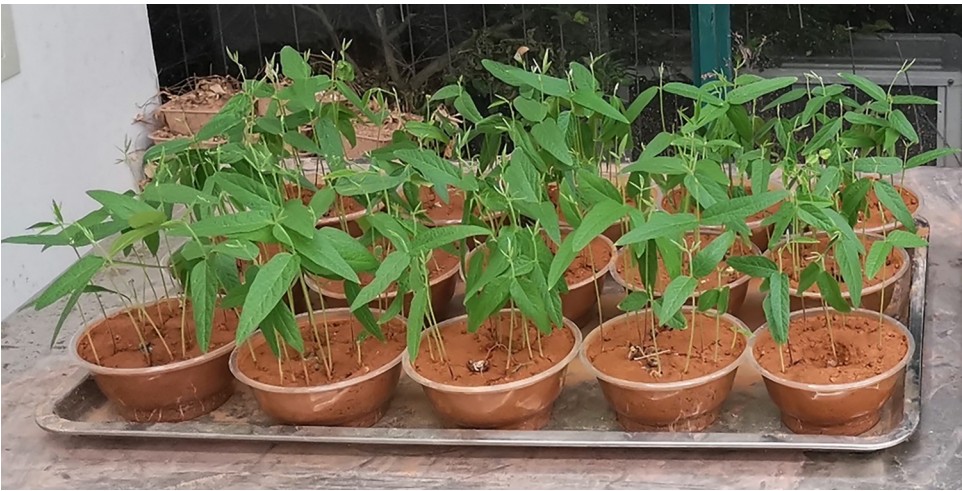

**Fig 16. Image showing the growth of wood bean plants.**

ring knife, and there was insufficient area for expansion after water absorption. As shown in Fig 20, soil particles, particularly weathered red-bed soil particles (with relatively small particle sizes), could only spread continuously in the inter-pore channels between the particles under

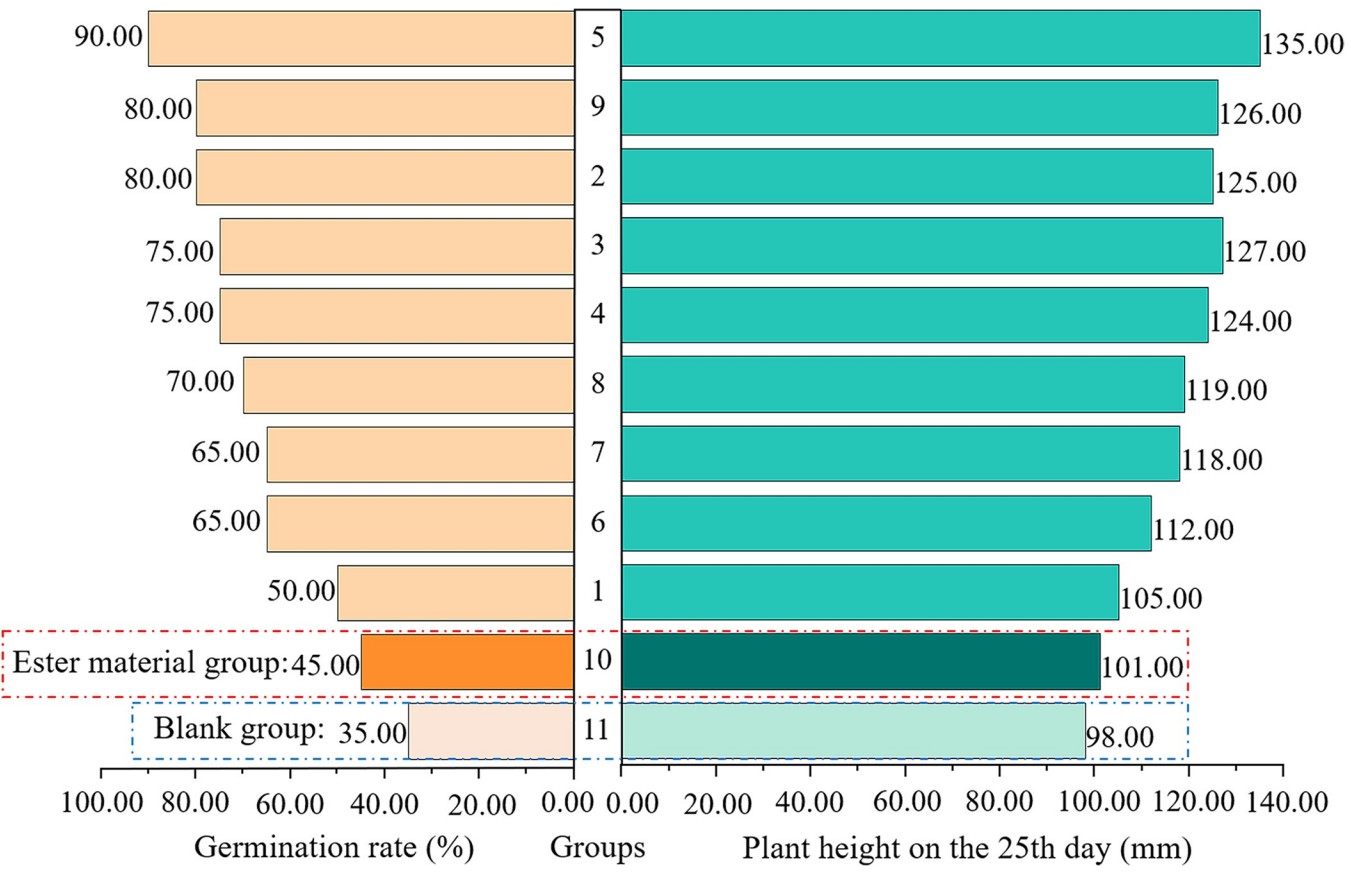

**Fig 17. Temporal change in plant growth in ester materials coupled with weathered red-bed soil.**

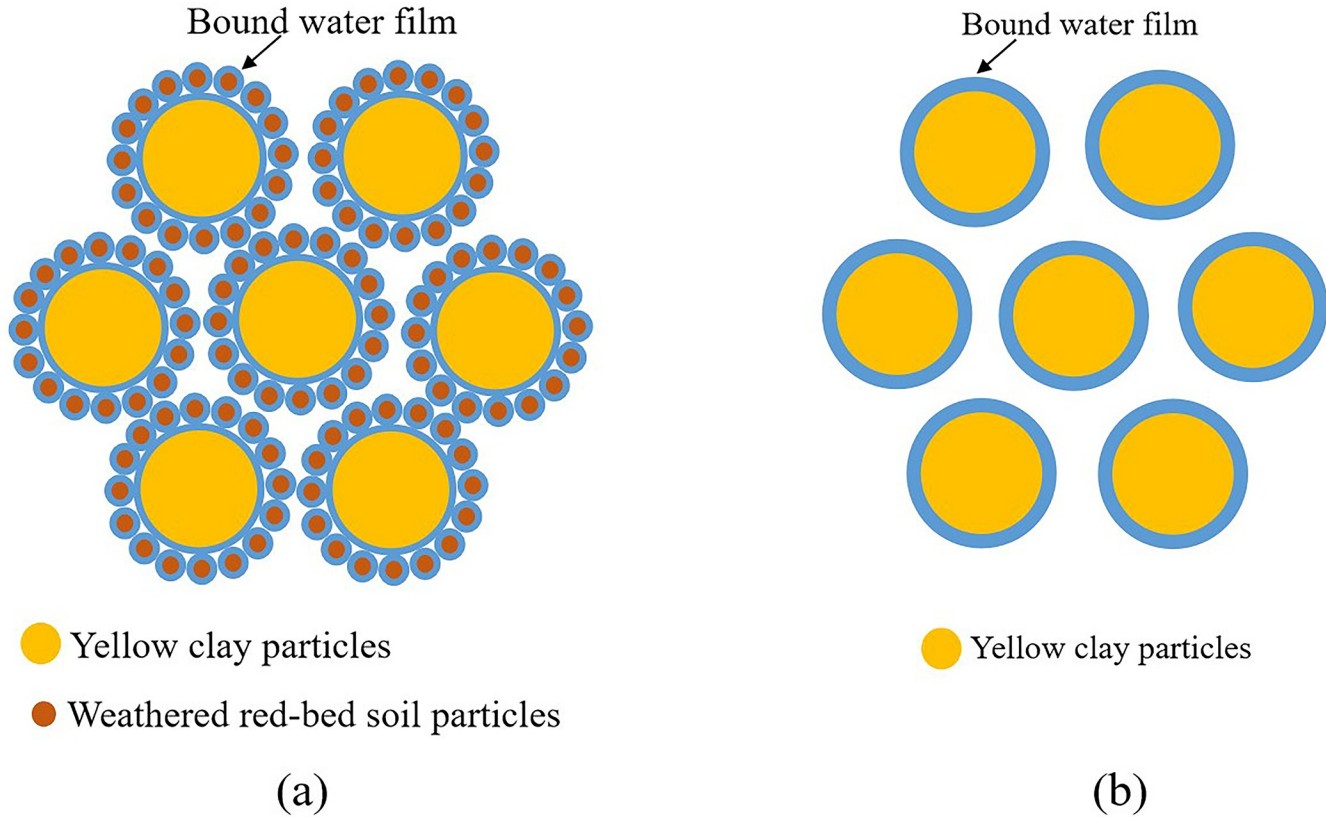

**Fig 18. Soil sample structure diagram with and without the addition of weathered red-bed soil.** (a) Soil sample structure without red-bed soil, (b) Soil sample structure of weathered red-bed soil.

seepage flow action, and gradually accumulated to block inter-pore connectivity. Therefore, in this case, the permeability coefficient of the soil sample decreased with the addition of weathered red-bed soil.

In summary, the interaction between weathered red-bed soil particles and water is primarily mediated by hydrogen bonding and surface forces, resulting in the adsorption of water molecules on the surface of the soil particles and the production of a hydration film. Therefore, the maximum amount of water that can be absorbed by the weathered red-bed soil is determined by its mineral composition, particle gradation, and pore properties. In addition, the interactions between lipids and water depend on the network structure created by the diffusion of molecular chains in water. This mesh structure can draw more water and limit evaporation owing to its permeability potential differences; however, it can provide a larger space for water bodies to fuse compared to that in weathered red-bed soil. Therefore, its ability to absorb and store water is greater than that of weathered red-bed soil, and its use can only minimize the need for fatty materials, rather than replace them.

## Conclusion

The aim of this study was to make effective use of weathered red-bed soils as ecological protection materials and reduce the use of artificial materials in soil erosion control projects, thereby reducing project costs and resource loss. Experiments on the coupling effect of weathered red-

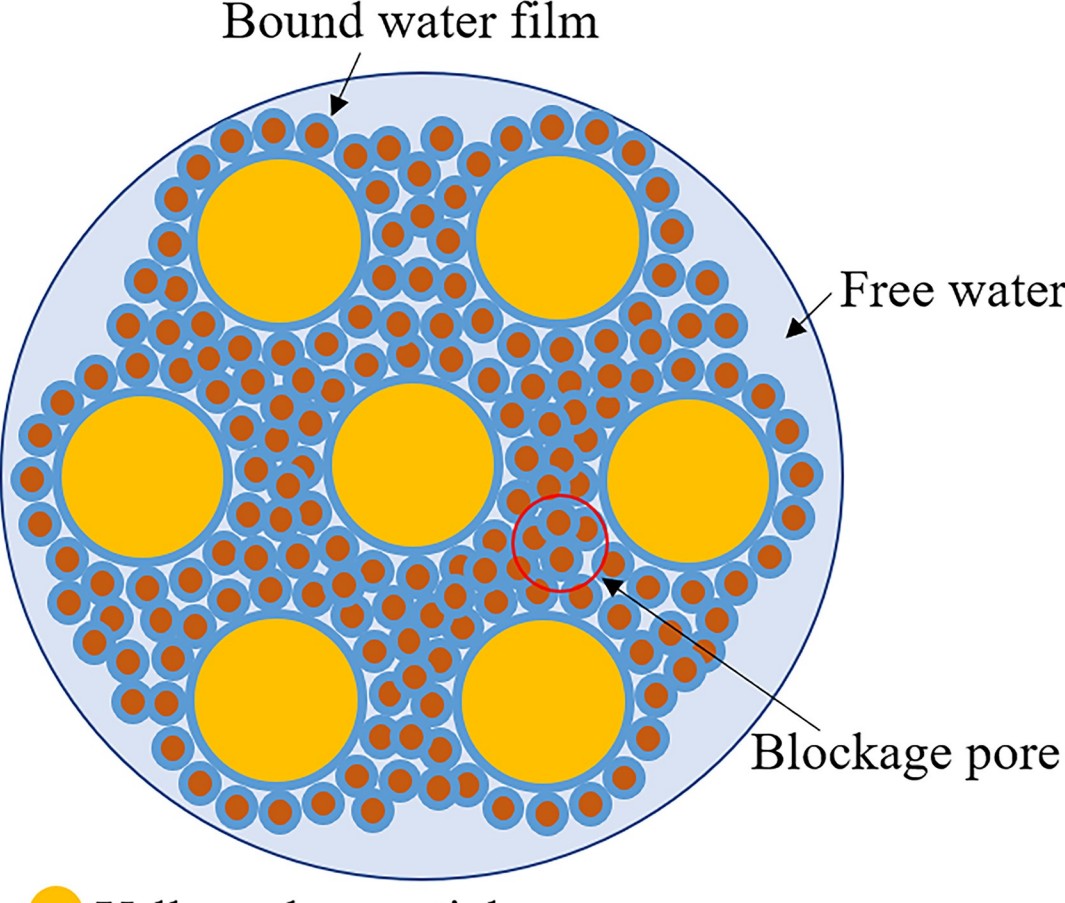

**Fig 19. Diagram showing the adsorption of cations in weathered red-bed soil particles.**

bed soils and ester materials were designed and performed to study the mechanism of action of weathered red-bed soils partially replacing ester materials, and the main conclusions are as follows:

(1) By studying the basic properties of weathered red-bed soils and ester materials, it was shown that weathered red-bed soil and ester materials have similar properties, including strong cementation, hydrophilicity, and expansibility. This provides new ideas and theoretical support for the study of the soil improvement effects of weathered red-bed soils partially replacing ester materials.

(2) Orthogonal tests were conducted on the coupled compounding of weathered red-bed soils and ester materials, as well as orthogonal tests of ecological effects. The results show that adding weathered red-bed soil to clay soil reduces erosion and shrinkage rates (by 15.5% and 5.4%, respectively) and increases water content, swelling rate, electrical conductivity, and permeability coefficient (by 9.3%, 3.6%, 118.0%, and 152.5%, respectively); moreover, it increases cohesion up to 31.9 kPa. Plant germination and height increased by 55% and 37 mm, respectively, at a ratio of 15 g/m$^2$ absorbent ester material, 2.5 g/m$^2$ adhesive ester material, and 5%

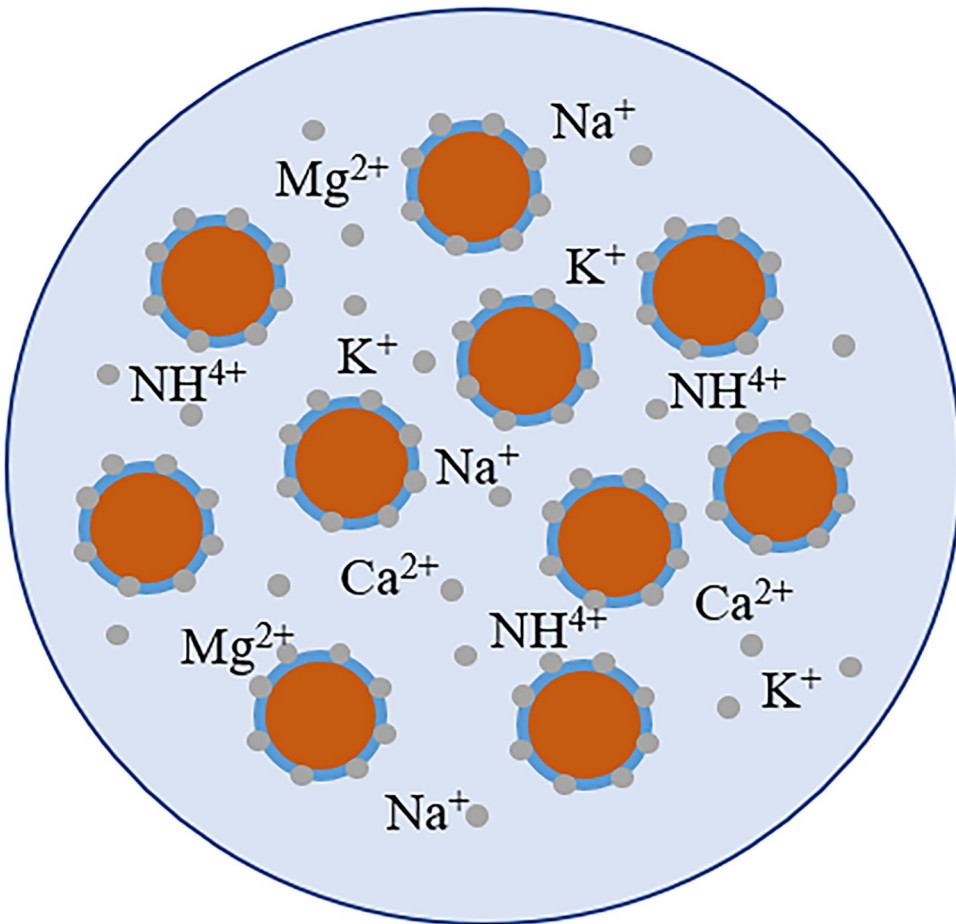

**Fig 20. Diagram showing the plug pores of yellow clay particles with the addition of weathered red-bed soil.**

weathered red-bed soil; the amount of ester material could be further reduced by 75%. Weathered red-bed soil effectively improved the soil structure and ensured long-term plant growth.

(3) The mechanism of the partial substitution of ester materials by weathered red-bed soils was thoroughly examined based on the mineral composition, soil structure, and electrical properties of these soils. This analysis serves as a foundation for the substitution of biological enzymes and polymer materials with comparable properties. Simultaneously, it offers a theoretical direction for utilizing weathered red-bed soils in other domains, such as infrastructure and industrial development.

## Supporting information

**S1 File.**
(DOCX)

## Author Contributions

**Conceptualization:** Zhen Liu.

**Data curation:** Cuiying Zhou.

**Formal analysis:** Jin Liao, Dexian Li.

**Funding acquisition:** Cuiying Zhou.

**Investigation:** Jin Liao, Cuiying Zhou.

**Methodology:** Yongtao Wu.

**Project administration:** Cuiying Zhou.

**Software:** Yongtao Wu.

**Supervision:** Zhen Liu, Cuiying Zhou.

**Validation:** Yongtao Wu, Jin Liao, Dexian Li.

**Visualization:** Yongtao Wu.

**Writing – original draft:** Yongtao Wu.

**Writing – review & editing:** Zhen Liu.

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
