## [Decision Letter · Decision Letter 0]

20 Nov 2023

PONE-D-23-32713The performance of partially substituted composite ester materials with weathered red-bed soil in ecological restorationPLOS ONE

Dear Dr. Zhou,

Thank you for submitting your manuscript to PLOS ONE. After careful consideration, we feel that it has merit but does not fully meet PLOS ONE’s publication criteria as it currently stands. Therefore, we invite you to submit a revised version of the manuscript that addresses the points raised during the review process.

We look forward to receiving your revised manuscript.

Kind regards,

Jianguo Wang, PhD

Academic Editor

PLOS ONE

Journal Requirements:

The research is supported by the National Natural Science Foundation of China (NSFC) (Grant Numbers: 42293354, 42293351, 42293355, 42277131, 41977230).

The research is supported by the National Natural Science Foundation of China (NSFC) (Grant Numbers: 42293354, 42293351, 42293355, 42277131, 41977230).

The research is supported by the National Natural Science Foundation of China (NSFC) (Grant Numbers: 42293354, 42293351, 42293355, 42277131, 41977230).

Reviewers' comments:

Reviewer's Responses to Questions

**Comments to the Author**

1. Is the manuscript technically sound, and do the data support the conclusions?

Reviewer #1: Yes

2. Has the statistical analysis been performed appropriately and rigorously? 

Reviewer #1: Yes

3. Have the authors made all data underlying the findings in their manuscript fully available?

Reviewer #1: Yes

4. Is the manuscript presented in an intelligible fashion and written in standard English?

Reviewer #1: Yes

5. Review Comments to the Author

Reviewer #1: This paper introduces an interesting approach of incorporating weathered red-bed soil as a partial alternative to composite ester materials for ecological restoration. The results show that the developed method has the potential to reduce the cost of ecological restoration by using specific ecological restoration indicators. This is an interesting study and paper can be accepted for publication after some revisions.

• The Abstract should be fortified by including specific quantitative results to provide a clearer snapshot of the findings.

• Within the Introduction, refrain from clustering multiple references at the end of a single sentence. Each cited study offers distinct insights; thus, summarizing the pivotal findings of each will guide readers through the evolution and direction of the research field. In the concluding portion of the Introduction, the authors must distinctly highlight the unique contributions and innovations of their research.

• While the authors embarked on a dual test of the composite properties and ecological impact of the weathered red-bed soil and ester materials, the rationale for this approach wasn't provided. Clarifying the purpose and significance of this test will ensure readers grasp its relevance.

• In the section discussing research content and methods, omit sentences that only serve as basic introductions.

• In the research content and methods section, the determination of the optimal ratio solely based on plant growth indicators from the ester material compounding experiment is limiting. To provide a more comprehensive insight, it would be prudent to consider all indicators listed in Table 1.

• Ensure that figure captions align with their respective mentions in the manuscript. Specifically, there's a discrepancy between Figure 9 and its corresponding reference in the text. Additionally, confirm the correctness of units and the layout of all tables and figures.

• Throughout the paper, it's observed that there's inconsistency in the presentation of significant figures and decimals (e.g., discrepancies between 1.52 and 1.21 in Table 1; 29.98 and 28.6 in Figure 10). Standardizing these figures will enhance the manuscript's precision.

• Figure 6, which delineates the cohesion variation in different ester materials with the inclusion of weathered red-bed soil, should be represented more accurately. Instead of mere curve connections, a fitting method that more closely mirrors the actual change would be more appropriate.

• In the Discussion, there's a palpable gap between the proposed mechanism and the derived experimental data. The proposed explanations lean heavily on inferred logic. To solidify the credibility of the proposed mechanism, drawing more overt correlations between the experimental data and the given interpretations is essential.

• In the Conclusion, initiate with a general statement before diving into point-wise findings.

6. PLOS authors have the option to publish the peer review history of their article (what does this mean?). If published, this will include your full peer review and any attached files.

Reviewer #1: No

---

## [Author Response · Author response to Decision Letter 0]

24 Nov 2023

Journal Requirements:

Point 1: When submitting your revision, we need you to address these additional requirements. Please ensure that your manuscript meets PLOS ONE's style requirements, including those for file naming. 

Response 1： Thank you for reminding me. We have modified the format of this article according to the template you provided.

Point 2: In your Methods section, please provide additional information regarding the permits you obtained for the work. Please ensure you have included the full name of the authority that approved the field site access and, if no permits were required, a brief statement explaining why.

Response 2: Thank you very much for your review of this article and your valuable comments. The test soil samples were taken from a slope site of Kaichun Expressway, Yangchun City, Guangdong Province. It is a slope ecological restoration project that we did and is approved for site access, citation [39] has been approved for site access.

Point 3: Thank you for stating in your Funding Statement: The research is supported by the National Natural Science Foundation of China (NSFC) (Grant Numbers: 42293354, 42293351, 42293355, 42277131, 41977230).

Response 3: We have stated in the cover letter as follows“This research is supported by the National Natural Science Foundation of China (Grant Numbers: 42293354, 42293351, 42293355, 42277131, 41977230). These fundings are all awarded by the corresponding author Cuiying Zhou. We declare

that the funder has no known competing financial interests or personal relationships that could have appeared to influence the work reported in this paper. The funders had no role in study design, data collection and analysis, decision to publish, or preparation of the manuscript.” 

Point 4: Thank you for stating the following in the Acknowledgments Section of your manuscript: 

The research is supported by the National Natural Science Foundation of China (NSFC) (Grant Numbers: 42293354, 42293351, 42293355, 42277131, 41977230).

The research is supported by the National Natural Science Foundation of China (NSFC) (Grant Numbers: 42293354, 42293351, 42293355, 42277131, 41977230).

Response 4: Thanks for reminding me. The funding information has been removed from the acknowledgements section of the manuscript and included in the cover letter.

Point 5: In your Data Availability statement, you have not specified where the minimal data set underlying the results described in your manuscript can be found. PLOS defines a study's minimal data set as the underlying data used to reach the conclusions drawn in the manuscript and any additional data required to replicate the reported study findings in their entirety. All PLOS journals require that the minimal data set be made fully available. For more information about our data policy, please see http://journals.plos.org/plosone/s/data-availability.

Response 5: Thank you for reminding me. Relevant data supporting the results of this study have been uploaded to a file named " ecological_restoration ".

Comments and Suggestions for Authors：

This paper introduces an interesting approach of incorporating weathered red-bed soil as a partial alternative to composite ester materials for ecological restoration. The results show that the developed method has the potential to reduce the cost of ecological restoration by using specific ecological restoration indicators. This is an interesting study and paper can be accepted for publication after some revisions.

Response: Thank you very much for your review of this article and your valuable comments. The author has modified the full text one by one according to your Suggestions. The specific modified content and the reply to the question are shown below.

Point 1: The Abstract should be fortified by including specific quantitative results to provide a clearer snapshot of the findings.

Response 1: Thank you very much for your review of this article and your valuable comments. We have added the quantitative results of the trial in the abstract section. You can see the supplementary content in L16-19.

Point 2: Within the Introduction, refrain from clustering multiple references at the end of a single sentence. Each cited study offers distinct insights; thus, summarizing the pivotal findings of each will guide readers through the evolution and direction of the research field. In the concluding portion of the Introduction, the authors must distinctly highlight the unique contributions and innovations of their research.

Response 2: Thank you very much for your review of this article and your valuable comments. We summarize the right and wrong of the works cited. The contribution of these works to this study is also analyzed and what corrections need to be made on this basis. You can see the supplementary content in L41-74.

Point 3: While the authors embarked on a dual test of the composite properties and ecological impact of the weathered red-bed soil and ester materials, the rationale for this approach wasn't provided. Clarifying the purpose and significance of this test will ensure readers grasp its relevance.

Response 3: Thank you very much for your review of this article and your valuable comments. We explain the purpose and significance of this test in the article. You can see the supplementary content in L146-149 and L243-245. And it also provides the rationale for this approach. You can see the supplementary content in L155 and L179.

Point 4: In the section discussing research content and methods, omit sentences that only serve as basic introductions. 

Response 4: Thank you very much for your review of this article and your valuable comments. We have checked the article text and omit sentences that only serve as basic introductions.

Point 5: In the research content and methods section, the determination of the optimal ratio solely based on plant growth indicators from the ester material compounding experiment is limiting. To provide a more comprehensive insight, it would be prudent to consider all indicators listed in Table 1.

Response 5: Thank you very much for your review of this article and your valuable comments. We have considered all the indicators in Table 1 in the ester material composite experiment You can see the supplementary content in L146-149 and Fig 4b.

Point 6: Ensure that figure captions align with their respective mentions in the manuscript. Specifically, there's a discrepancy between Figure 9 and its corresponding reference in the text. Additionally, confirm the correctness of units and the layout of all tables and figures.

Response 6: Thank you very much for your review of this article and your valuable comments. 

We have mapped the figure captions to the respective mentions in the manuscript and confirmed the correctness of the units and the layout of all tables and figures. You can see the supplement content in the full text.

Point 7: Throughout the paper, it's observed that there's inconsistency in the presentation of significant figures and decimals (e.g., discrepancies between 1.52 and 1.21 in Table 1; 29.98 and 28.6 in Figure 10). Standardizing these figures will enhance the manuscript's precision.

Response 7: Thank you very much for your review of this article and your valuable comments. We apologize for not noticing this. We have standardized the significant figures throughout the paper. You can see the revisions in the text.

Point 8: Figure 6, which delineates the cohesion variation in different ester materials with the inclusion of weathered red-bed soil, should be represented more accurately. Instead of mere curve connections, a fitting method that more closely mirrors the actual change would be more appropriate.

Response 8: Thank you very much for your review of this article and your valuable comments. We have used Modified Bezier curve fitting in Figure 6, and the fitting equation can be seen in Figure 6.

Point 9: In the Discussion, there's a palpable gap between the proposed mechanism and the derived experimental data. The proposed explanations lean heavily on inferred logic. To solidify the credibility of the proposed mechanism, drawing more overt correlations between the experimental data and the given interpretations is essential.

Response 9: Thank you very much for your review of this article and your valuable comments. We have linked the correlation between the proposed mechanism and the derived experimental data. You can see the supplementary content in L381-420.

Point 10: In the Conclusion, initiate with a general statement before diving into point-wise findings.

 Response 10: Thank you very much for your review of this article and your valuable comments. We have started with a general statement, followed by a point-by-point conclusion. You can see the supplementary content in L462-487.

---

## [Decision Letter · Decision Letter 1]

26 Dec 2023

PONE-D-23-32713R1The performance of partially substituted composite ester materials with weathered red-bed soil in ecological restorationPLOS ONE

Dear Dr. Zhou,

Thank you for submitting your manuscript to PLOS ONE. After careful consideration, we feel that it has merit but does not fully meet PLOS ONE’s publication criteria as it currently stands. Therefore, we invite you to submit a revised version of the manuscript that addresses the points raised during the review process.

**ACADEMIC EDITOR: **

Please carefully address the comments from reviewers and improve the quality of the manuscript.  

We look forward to receiving your revised manuscript.

Kind regards,

Jianguo Wang, PhD

Academic Editor

PLOS ONE

Reviewers' comments:

Reviewer's Responses to Questions

**Comments to the Author**

1. If the authors have adequately addressed your comments raised in a previous round of review and you feel that this manuscript is now acceptable for publication, you may indicate that here to bypass the “Comments to the Author” section, enter your conflict of interest statement in the “Confidential to Editor” section, and submit your "Accept" recommendation.

Reviewer #2: (No Response)

2. Is the manuscript technically sound, and do the data support the conclusions?

Reviewer #2: No

3. Has the statistical analysis been performed appropriately and rigorously? 

Reviewer #2: N/A

4. Have the authors made all data underlying the findings in their manuscript fully available?

Reviewer #2: Yes

5. Is the manuscript presented in an intelligible fashion and written in standard English?

Reviewer #2: Yes

6. Review Comments to the Author

Reviewer #2: The manuscript deals with the investigation of the performance of partially substituted composite ester materials with weathered red bed soil in ecological restoration. The entire manuscript is recommended for a careful and thorough check by a certified proofreading agency to ensure that the revised article is free from grammatical mistakes, spelling mistakes, punctuation errors, etc. The authors have to place the superscripts and subscripts throughout the content correctly. The authors are expected to address the following shortcomings and revise their manuscript by providing valid reasons for each of the queries raised.

1. Abstract: In line 20, the authors can explicitly describe the kind of modification endured by the soil structure.

2. The lines, “the focus in ecological restoration is environmental degradation” in lines 33 and 34 can be avoided as the same has been stated in the first sentence itself. A suitable reference can be cited for the statement describing the effectiveness of ester materials in line 35. It would also be appropriate to mention some ester materials which has proved their efficacy in the same statement.

3. How was the viscosity of the red bed soil determined in the current study? Mention that under methodology. Also, mention the soil classification.

4. What exactly is the synthetic glue referred to in line 160?

5. Referring to Table 3, how are the various comparative statements regarding the properties validated for ester materials and red bed soil?

6. Specifically name the ester material used in the current study as there are physical, chemical, and biological materials used for the same purpose. Since the ester materials were sprayed onto the soil mixture, what is the solubility of these materials in water?

7. From the plant growth line steps listed in line 157, it is understood that there are two soil types used for developing a preliminary soil mix. What is the purpose of taking yellow-brown clay collected from highway sites? Also, the physical properties of red bed soil have not been mentioned in Table 2.

8. In line 185, is there any code followed for determining the scouring resistance? How were the tilt angle and flow rate fixed?

9. In lines 237-239, Why are the results of the variable head test given under the procedure section?

10. In Table 6, provide as a footnote the significance of the notations p, q, and r. The same comment applies to Table 7 also.

11. What is the role played by ester materials in decreasing the erosion rate? How did the hydrophilicity of ester materials and red bed soil affect the phenomenon?

12. If yellow soil is the base soil to which the ester materials and red soil are added, then where are the properties of the untreated yellow soil such as permeability coefficient, erosion rate, shrinkage, etc?

13. In the mechanism, it would be appropriate to discuss the mechanisms related to each property variation in the same order of properties as followed in the previous section.

14. In the entire mechanism explained for the mechanical properties, the role of ester materials and their interaction with the yellow soil has not been elaborated. If the ester materials do not contribute to the improvement in any way, then the purpose of highlighting their properties in Table 3 is defeated.

15. The article fails to explain the synergistic interaction between red bed soil, base soil, and ester material. This aspect needs to be highlighted or certain justifications need to be brought for eliminating these discussions.

16. In the supplementary file, the tables are mentioned as figures. The figures are not made available within the file.

17. The font in Figs. 1a and 1b are not clear and hence need to be replaced. The same comment applies to Figs. 4a, 5, 6, 7, 8, 9, 10, 11, 13, 14a and b, 16. The x-axis of Fig. 1b needs to be written as particle size. There is an issue with the figure numbers within the running text, figure titles, and supplementary files.

18. It is not necessary to mention the values in the trend line in Figs. 5 and 6. There is an overlap of legends in Figs. 6 and 14b.

7. PLOS authors have the option to publish the peer review history of their article (what does this mean?). If published, this will include your full peer review and any attached files.

Reviewer #2: **Yes: **Prof Arif Ali Baig Moghal

---

## [Author Response · Author response to Decision Letter 1]

12 Jan 2024

Comments and Suggestions for Authors：

The manuscript deals with the investigation of the performance of partially substituted composite ester materials with weathered red bed soil in ecological restoration. The entire manuscript is recommended for a careful and thorough check by a certified proofreading agency to ensure that the revised article is free from grammatical mistakes, spelling mistakes, punctuation errors, etc. The authors have to place the superscripts and subscripts throughout the content correctly. The authors are expected to address the following shortcomings and revise their manuscript by providing valid reasons for each of the queries raised.

Response: Thank you very much for your review of this article and your valuable comments. The author has modified the full text one by one according to your Suggestions. The specific modified content and the reply to the question are shown below.

Point 1: Abstract: In line 20, the authors can explicitly describe the kind of modification endured by the soil structure.

Response 1: Thank you very much for your review of this article and your valuable comments. We have added the type of alteration to which the soil structure is subjected in the abstract section. You can see the supplementary content in L17-19.

Point 2: The lines, “the focus in ecological restoration is environmental degradation” in lines 33 and 34 can be avoided as the same has been stated in the first sentence itself. A suitable reference can be cited for the statement describing the effectiveness of ester materials in line 35. It would also be appropriate to mention some ester materials which has proved their efficacy in the same statement.

Response 2: Thank you very much for your review of this article and your valuable comments. We have deleted the phrase " the focus in ecological restoration is environmental degradation " in lines 33 and 34, and cited suitable references for the statement of the effectiveness of the ester material in line 35. You can see the supplementary content in L33.

Point 3: How was the viscosity of the red bed soil determined in the current study? Mention that under methodology. Also, mention the soil classification.

Response 3: Thank you very much for your review of this article and your valuable comments. We have supplemented the red-layer weathered soil measurements in our research methodology. That is, ① Grinding. 2Kg of red-layered weathered soil was cracked with a wooden hammer to disperse it. ② Drying. The red-layered weathered soil was dried in an oven at 105°C for 24 h. It was removed and placed in a desiccator to cool to room temperature. ③ Sieving. Sieve the red layer of weathered soil using a standard sieve with apertures of 5, 2, 1, 0.5, 0.25, 0.1 and 0.075 in turn. ④ Mixing. Take the same mass of red layer weathered soil, calculate the amount of water added according to its average natural water content, add water and mix it to make slurry. ⑤ Measurement. Use a viscometer to measure the viscosity of the red layer weathered soil. You can see the supplementary content in L106 and L108.

Point 4: What exactly is the synthetic glue referred to in line 160?

Response 4: Thank you very much for your review of this article and your valuable comments. The synthetic glue in line 160 refers to adhesive ester materials, as described in " Characterization of the ecological restoration effect of composite ester materials ". We have standardized the names throughout the text.

Point 5: Referring to Table 3, how are the various comparative statements regarding the properties validated for ester materials and red bed soil?

Response 5: Thank you very much for your review of this article and your valuable comments. We quoted a comparative verification of various properties by citing previous research results on the properties of ester materials and red bed soil. You can see the supplementary content in Table 3.

Point 6: Specifically name the ester material used in the current study as there are physical, chemical, and biological materials used for the same purpose. Since the ester materials were sprayed onto the soil mixture, what is the solubility of these materials in water?

Response 6: Thank you very much for your review of this article and your valuable comments. We have added a specific description of the ester materials used in the current study in " Characterization of the ecological restoration effect of composite ester materials". Binding ester materials are insoluble in water, but are well dispersed in water. You can see the supplementary content in L92.

Point 7: From the plant growth line steps listed in line 157, it is understood that there are two soil types used for developing a preliminary soil mix. What is the purpose of taking yellow-brown clay collected from highway sites? Also, the physical properties of red bed soil have not been mentioned in Table 2.

Response 7: Thank you very much for your review of this article and your valuable comments. We have added the purpose and the physical properties of the red-layer weathered clay in the text. The purpose of the yellow-brown clay collected from the highway site was the poor ecological effect of the area and the fact that the soil was typical of yellow clay in southern China. You can see the supplementary content in L146-148 and Table 2.

Point 8: In line 185, is there any code followed for determining the scouring resistance? How were the tilt angle and flow rate fixed?

Response 8: Thank you very much for your review of this article and your valuable comments. The tilt angle and flow rate of the scour are determined based on the actual project slope gradient as well as the local rainfall characteristics. We have added this in the text. You can see the supplementary content in L190-192.

Point 9: In lines 237-239, Why are the results of the variable head test given under the procedure section?

Response 9: Thank you very much for your review of this article and your valuable comments. The results of the variable head test were used to calculate the permeability coefficient. We have added details in the text. You can see the supplementary content in L228-230.

Point 10: In Table 6, provide as a footnote the significance of the notations p, q, and r. The same comment applies to Table 7 also.

Response 10: Thank you very much for your review of this article and your valuable comments. We have added footnotes to Tables 6 and 7 to provide the significance of the notations p (P), a (A) and r (R). You can see the supplementary content in L269 and L281.

Point 11: What is the role played by ester materials in decreasing the erosion rate? How did the hydrophilicity of ester materials and red bed soil affect the phenomenon?

Response 11: Thank you very much for your review of this article and your valuable comments. The polymer chain of ester materials contains active groups, they link with each other and soil particles via hydrogen bonding, intermolecular force, and ion exchange, and the polymer chain entangles the loose soil particles into a whole. As a result, there is an increase in the soil's ability to withstand gravity and rainwater scouring damage, that is, scouring resistance. We've added to the text. You can see the supplementary content in L414-416. The hydrophilicity of the ester material and the red-layered soil affects the structure of the soil sample by changing the thickness of the hydrated film, which acts to reduce erosion. You can see the addition in Fig. 18.

Point 12: If yellow soil is the base soil to which the ester materials and red soil are added, then where are the properties of the untreated yellow soil such as permeability coefficient, erosion rate, shrinkage, etc?

Response 12: Thank you very much for your review of this article and your valuable comments. The untreated yellow soil in the ester material compounding test is group 10, and the untreated yellow soil's in the red layer weathered soil coupled with ester material compounding test and ecological effect test is group 1. Corresponding properties such as permeability coefficient, erosion rate and shrinkage are in the result analysis.

Point 13: In the mechanism, it would be appropriate to discuss the mechanisms related to each property variation in the same order of properties as followed in the previous section.

Response 13: Thank you very much for your review of this article and your valuable comments. We have discussed the mechanisms associated with each property change in the same property order followed in the previous section in Mechanisms. You can see the supplementary content in L395-469.

Point 14: In the entire mechanism explained for the mechanical properties, the role of ester materials and their interaction with the yellow soil has not been elaborated. If the ester materials do not contribute to the improvement in any way, then the purpose of highlighting their properties in Table 3 is defeated.

Response 14: Thank you very much for your review of this article and your valuable comments. Our group has conducted preliminary research and practical engineering experience on the role of ester materials and their interaction with the yellow clay, and the results of published papers indicate that ester materials are effective in the remediation of yellow clay. We have cited relevant references in the article. You can see the supplementary content in L94-99.

Point 15: The article fails to explain the synergistic interaction between red bed soil, base soil, and ester material. This aspect needs to be highlighted or certain justifications need to be brought for eliminating these discussions.

Response 15: Thank you very much for your review of this article and your valuable comments. We have added to the discussion the mechanism of coupled action of yellow clays and red bed soil with ester materials. You can see the supplementary content in L395-403.

Point 16: In the supplementary file, the tables are mentioned as figures. The figures are not made available within the file.

Response 16: Thank you very much for your review of this article and your valuable comments. We have added tables and figures to the supplementary file. You can see them in the supplementary file.

Point 17: The font in Figs. 1a and 1b are not clear and hence need to be replaced. The same comment applies to Figs. 4a, 5, 6, 7, 8, 9, 10, 11, 13, 14a and b, 16. The x-axis of Fig. 1b needs to be written as particle size. There is an issue with the figure numbers within the running text, figure titles, and supplementary files.

Response 17: Thank you very much for your review of this article and your valuable comments. We have replaced the photos with unclear fonts in the tex. You can see the changes in the text.

Point 18: It is not necessary to mention the values in the trend line in Figs. 5 and 6. There is an overlap of legends in Figs. 6 and 14b.

Response 18: Thank you very much for your review of this article and your valuable comments. We have removed the values from the trend lines in Figs 5 and 6 and modified the legends in Figs 6 and 14b. You can see the supplementary content in Fig. 6, Figs 7 and 15b

---

## [Decision Letter · Decision Letter 2]

18 Jan 2024

PONE-D-23-32713R2The performance of partially substituted composite ester materials with weathered red-bed soil in ecological restorationPLOS ONE

Dear Dr. Zhou,

Thank you for submitting your manuscript to PLOS ONE. After careful consideration, we feel that it has merit but does not fully meet PLOS ONE’s publication criteria as it currently stands. Therefore, we invite you to submit a revised version of the manuscript that addresses the points raised during the review process.

We look forward to receiving your revised manuscript.

Kind regards,

Jianguo Wang, PhD

Academic Editor

PLOS ONE

Reviewers' comments:

Reviewer's Responses to Questions

**Comments to the Author**

1. If the authors have adequately addressed your comments raised in a previous round of review and you feel that this manuscript is now acceptable for publication, you may indicate that here to bypass the “Comments to the Author” section, enter your conflict of interest statement in the “Confidential to Editor” section, and submit your "Accept" recommendation.

Reviewer #2: (No Response)

2. Is the manuscript technically sound, and do the data support the conclusions?

Reviewer #2: Partly

3. Has the statistical analysis been performed appropriately and rigorously? 

Reviewer #2: Yes

4. Have the authors made all data underlying the findings in their manuscript fully available?

Reviewer #2: Yes

5. Is the manuscript presented in an intelligible fashion and written in standard English?

Reviewer #2: Yes

6. Review Comments to the Author

Reviewer #2: The manuscript deals with the performance of partially substituted composite ester materials with weathered red bed soil in ecological restoration. Referring to the revised manuscript and comments provided by the authors, the following changes have been recommended. The article can be considered for publication after making the following changes.

1. The entire manuscript is recommended for a careful and thorough check by a certified proofreading agency to ensure that the revised article is free from grammatical mistakes, spelling mistakes, punctuation errors, etc. The authors have to place the superscripts and subscripts throughout the content correctly.

2. The sentence in lines 17-19 does not provide clarity and the authors are requested to modify it as “The experimental findings indicated that the soil modified using ester materials exhibited an improvement in strength, water retention, and aeration due to change in the soil structure”.

3. Introduction: The authors haven’t mentioned ester materials previously used for soil improvement. Rather, only the references have been cited.

4. In Table 3, the authors can cite the relevant references against each point rather than citing them against the title.

5. Figures 2, 9, and 12 need to be replaced with better-quality images.

7. PLOS authors have the option to publish the peer review history of their article (what does this mean?). If published, this will include your full peer review and any attached files.

Reviewer #2: **Yes: **Prof Arif Ali Baig Moghal

---

## [Author Response · Author response to Decision Letter 2]

27 Jan 2024

Comments and Suggestions for Authors：

The manuscript deals with the performance of partially substituted composite ester materials with weathered red bed soil in ecological restoration. Referring to the revised manuscript and comments provided by the authors, the following changes have been recommended. The article can be considered for publication after making the following changes.

Response: Thank you very much for your review of this article and your valuable comments. The author has modified the full text one by one according to your Suggestions. The specific modified content and the reply to the question are shown below.

Point 1: The entire manuscript is recommended for a careful and thorough check by a certified proofreading agency to ensure that the revised article is free from grammatical mistakes, spelling mistakes, punctuation errors, etc. The authors have to place the superscripts and subscripts throughout the content correctly.

Response 1: We have polished the entire manuscript in the editing agency and corrected grammar, spelling, punctuation, and superscript and subscript issues throughout the manuscript. You can see the corrections in the manuscript.

Point 2: The sentence in lines 17-19 does not provide clarity and the authors are requested to modify it as “The experimental findings indicated that the soil modified using ester materials exhibited an improvement in strength, water retention, and aeration due to change in the soil structure”.

Response 2: We have revised the sentence from lines 17 to 19 to modify: "Experimental results show that due to changes in soil structure, soil modified with ester materials exhibits improvements in strength, water retention, and aeration.". You can see the supplementary content in L17-19.

Point 3: Introduction: The authors haven’t mentioned ester materials previously used for soil improvement. Rather, only the references have been cited.

Response 3: We have summarized the ester materials previously used for soil improvement and cited corresponding references in the introduction. You can see the supplementary content in L32-33.

Point 4: In Table 3, the authors can cite the relevant references against each point rather than citing them against the title.

Response 4: We have cited relevant references for each point in Table 3. You can see the supplementary content in Table 3.

Point 5: Figures 2, 9, and 12 need to be replaced with better-quality images.

Response 5: We have replaced Figures 2, 9, and 12 with better quality images. You can see the supplementary content in Figures 2, 9, and 12.

---

## [Decision Letter · Decision Letter 3]

9 Feb 2024

The performance of partially substituted composite ester materials with weathered red-bed soil in ecological restoration

PONE-D-23-32713R3

Dear Dr. Zhou,

We’re pleased to inform you that your manuscript has been judged scientifically suitable for publication and will be formally accepted for publication once it meets all outstanding technical requirements.

Kind regards,

Jianguo Wang, PhD

Academic Editor

PLOS ONE

Additional Editor Comments (optional):

Reviewers' comments:

Reviewer's Responses to Questions

**Comments to the Author**

1. If the authors have adequately addressed your comments raised in a previous round of review and you feel that this manuscript is now acceptable for publication, you may indicate that here to bypass the “Comments to the Author” section, enter your conflict of interest statement in the “Confidential to Editor” section, and submit your "Accept" recommendation.

Reviewer #2: All comments have been addressed

2. Is the manuscript technically sound, and do the data support the conclusions?

Reviewer #2: Yes

3. Has the statistical analysis been performed appropriately and rigorously? 

Reviewer #2: N/A

4. Have the authors made all data underlying the findings in their manuscript fully available?

Reviewer #2: Yes

5. Is the manuscript presented in an intelligible fashion and written in standard English?

Reviewer #2: Yes

6. Review Comments to the Author

Reviewer #2: The authors have addressed my comments raised during the first review. The reviewer is satisfied with their response and recommends acceptance of the revised version of the manuscript.

7. PLOS authors have the option to publish the peer review history of their article (what does this mean?). If published, this will include your full peer review and any attached files.

Reviewer #2: **Yes: **Prof Arif Ali Baig Moghal

---

## [Editor Report · Acceptance letter]

25 Mar 2024

PONE-D-23-32713R3 

PLOS ONE

Dear Dr. Zhou, 

I'm pleased to inform you that your manuscript has been deemed suitable for publication in PLOS ONE. Congratulations! Your manuscript is now being handed over to our production team.

Kind regards, 

on behalf of

Dr. Jianguo Wang 

Academic Editor

PLOS ONE